

# Exotic invertible phases with higher-group symmetries

Po-Shen Hsin[1], Wenjie Ji[2] and Chao-Ming Jian[3]

**1** Walter Burke Institute for Theoretical Physics, California Institute of Technology, Pasadena, CA 91125, USA
**2** Department of Physics, University of California, Santa Barbara, CA 93106, USA
**3** Department of Physics, Cornell University, Ithaca, New York 14853, USA

## Abstract

We investigate a family of invertible phases of matter with higher-dimensional exotic excitations in even spacetime dimensions, which includes and generalizes the Kitaev's chain in 1+1d. The excitation has $\mathbb{Z}_2$ higher-form symmetry that mixes with the spacetime Lorentz symmetry to form a higher group spacetime symmetry. We focus on the invertible exotic loop topological phase in 3+1d. This invertible phase is protected by the $\mathbb{Z}_2$ one-form symmetry and the time-reversal symmetry, and has surface thermal Hall conductance not realized in conventional time-reversal symmetric ordinary bosonic systems without local fermion particles and the exotic loops. We describe a UV realization of the invertible exotic loop topological order using the $SO(3)_-$ gauge theory with unit discrete theta parameter, which enjoys the same spacetime two-group symmetry. We discuss several applications including the analogue of "fermionization" for ordinary bosonic theories with $\mathbb{Z}_2$ non-anomalous internal higher-form symmetry and time-reversal symmetry.



# 1 Introduction

Invertible or short-range entangled phases of matter have a unique ground state with an energy gap. They are believed to be classified by global symmetry [1–3]. They have important applications such as topological insulators and topological superconductors, see *e.g.* [4–8]. They can be characterized by their boundary properties, which realize the global symmetry in an anomalous way as specified by the invertible phase in the bulk. If the symmetry in invertible phases is gauged, their effective actions are described by theta terms in gauge theories, which have important consequences for the dynamics, see *e.g.* [9,10]. In particular, there is a single invertible 1+1d fermionic phase protected by fermion parity symmetry. On the boundary, the phase hosts the famous Majorana zero-mode. [11]

In this work, we investigate a family of invertible phases in arbitrary even spacetime dimensions, which includes the non-trivial phase of the Kitaev's chain in 1+1d [11,12]. We will focus on the invertible phase in 3+1d, which we call the "invertible exotic loop topological order (iELTO)". The theory has a unique loop excitation. When the exotic loops cross each other *i.e.* when their open worldsheet intersects in spacetime, the process produces a minus sign in the correlation function.[1] The invertible exotic loop topological order has *spacetime two-group global symmetry* that combines the $O(4)$ spacetime Lorentz symmetry, which includes the time-reversal symmetry, and $\mathbb{Z}_2$ one-form symmetry.[2] We review some general properties of higher-group symmetries in Section A. The exotic loop excitation transforms under the $\mathbb{Z}_2$ one-form symmetry.[3] The spacetime two-group symmetry implies that the exotic loops obey several properties: for instance, the Lorentz symmetry acts in an anomalous way on the exotic loop. The exotic loop invertible phase in 3+1d has several interesting boundary physics: (1) the boundary has an anti-semion, which is similar to the dangling Majorana fermion edge modes in Kitaev's chain [11,17]. (2) the boundary has a chiral central charge that is different from conventional time-reversal symmetric ordinary bosonic systems *i.e.* systems that consist of boson particles but not exotic loops. The boundary chiral central charge of the exotic loop invertible phase is indicative of a finite thermal Hall conductance on the boundary. A strict 2+1d time-reversal symmetric system has a vanishing thermal Hall conductance, while a system that resides on the boundary of a 3+1d bulk bosonic time-reversal invariant invertible phase can have thermal Hall conductance that equals an integer multiple of four [18] in units of $\frac{\pi k_B^2 T}{6\hbar}$ with $T$ being the temperature. In contrast, the invertible exotic loop topological order phase has a boundary thermal Hall conductance that equals $\pm 1$, which distinguishes the exotic loop phase from ordinary time-reversal symmetric bosonic systems. The thermal Hall conductance on the boundary is a measurable prediction for the invertible exotic loop phase.

We propose several gauge theory models for the invertible exotic loop phase. The first is described in terms of a $\mathbb{Z}_2$ two-form gauge theory. The second model is the $SO(3)$ gauge theory with $\theta = 2\pi$, which also has the two-group spacetime symmetry and the exotic loop excitation. Moreover, by coupling the gauge theory to matter fields, we find a theory with exotic loops, describing a plausible deconfined quantum critical point with the spacetime two-group symmetry which includes the $\mathbb{Z}_2$ one-form symmetry and time-reversal symmetry.

The results can be generalized to any even spacetime dimensions. The invertible phase in $2n$ spacetime dimensions has a $\mathbb{Z}_2$ $(n-1)$-form symmetry that extends the spacetime rotation group $O(2n)$ to a *spacetime n-group symmetry*. The theory has a unique excitation, the $(n-1)$-dimensional membrane. When the $(n-1)$-membranes cross *i.e.* their world history intersect in spacetime, the process produces a minus sign. The spacetime $n$-group symmetry implies that the $(n-1)$-dimensional membranes, which are charged under the $\mathbb{Z}_2$ $(n-1)$-form symmetry, obey properties similar to the exotic loop in the iELTO phase. When $n$ is even, breaking the time-reversal symmetry reduces the spacetime $n$-group symmetry to the direct product of $\mathbb{Z}_2$ $(n-1)$-form symmetry and ordinary Lorentz symmetry $SO(2n)$, and the invertible phase reduces to an invertible phase with $\mathbb{Z}_2$ $(n-1)$-form symmetry. Thus the time-reversal symmetry is essential to distinguish these invertible phases from other invertible phases discussed in the literature.

---

[1]This is reminiscent of the fermion particle exchange statistics in 1+1d.

[2]The higher-form symmetries in this work are the relativistic higher-form symmetries [13], where the generators can be continuously deformed without changing their eigenvalues unless crossing a charged operator. In the terminology of [14], they are "unfaithful higher-form symmetries", which means that small deformations of the symmetry generator do not change its value, in contrast to "faithful higher-form symmetries" where the symmetry generators supported on different submanifolds are different, such as the higher-form symmetries in the Toric code lattice model [15].

[3]This is similar to that the symmetry group for fermionic particles is $Spin(d)$ in $d$ spacetime dimension, which is the Lorentz symmetry $SO(d)$ extended by the $\mathbb{Z}_2$ fermion parity symmetry, see *e.g.* [16].

## 1.1 Organization

The note is organized as follows. In Section 2 we review some properties of ordinary fermionic phases in 1+1d and an effective $\mathbb{Z}_2$ gauge theory description for the Kitaev's chain. In Section 3 we discuss a phase with exotic loops in 3+1d protected by a spacetime two-group symmetry that combines $\mathbb{Z}_2$ one-form symmetry and time-reversal symmetry. In Section 4 we generalize the result to higher dimensions. In Section 5 we discuss an application to a generalization of "fermionizations" procedure. In Section 6 we comment on some future directions.

There are several appendices. In Appendix A we review some properties of higher-group symmetry. In Appendix B we summarized some mathematical properties of quadratic functions used to define the gauge theories that describe the invertible fermionic phases. In Appendix C we give some examples of the partition function for iELTO phase on simple manifolds. In Appendix D we discussed the continuum description for the invertible exotic loop phase in 3+1d using $U(1)$ two-form gauge theories, when the spacetime is orientable. In Appendix E we give some example of "fermionizations" using the exotic invertible phases discussed in Section 3 and 4.

## 2 Review: invertible fermionic particle topological order in 1+1d

We begin with a review for some properties of the invertible fermionic topological order in 1+1d [11]. We first describe the extension of the Lorentz group by the fermion parity symmetry and the corresponding background field in a Lorentz-invariant formulation of invertible fermionic topological orders. Then, we describe the phase introduced in Ref. [11] using an invertible $\mathbb{Z}_2$ gauge theory coupled to the background arising from the extension of the Lorentz group. In next section, we make the generalization to invertible phases protected by higher form symmetry and spacetime symmetry, which together form a higher-group.

### 2.1 Fermion parity symmetry and properties of fermion particles

We will review several implications of the fermion parity symmetry in invertible fermionic topological orders. These properties will have analogue for the higher-group symmetry discussed in later sections.

The spacetime symmetry $Spin(d)$ is a group extension of the Lorentz group $SO(d)$ by the $\mathbb{Z}_2$ fermionic parity that transforms fermionic particles[4]

$$1 \to \mathbb{Z}_2 \to Spin(d) \to SO(d) \to 1 \,. \tag{1}$$

The extension is specified by the second Stiefel Whitney class $w_2 \in H^2(SO(d), \mathbb{Z}_2)$ [19]. This means that the symmetry actions $R_h$ of $h \in SO(d)$ and $a \in \mathbb{Z}_2$ obeys the following algebra

$$R_h R_{h'} = a^{w_2(h,h')} R_{hh'} \,, \tag{2}$$

where $a^{w_2(h,h')}$ with $w_2(h,h') = 0, 1$ acts on bosonic particles (neutral under $\mathbb{Z}_2$, $a = +1$) by $+1$ and on fermionic particles (charged under $\mathbb{Z}_2$, $a = -1$) by $(-1)^{w_2(h,h')}$. In other words, the fermionic particle transforms as a projective representation of $SO(d)$.

The algebraic relation implies that, on a spacetime manifold $M$, the background $\rho_1$, which is one-form gauge field, for the $\mathbb{Z}_2$ fermion parity symmetry and the background $A$, another

---

[4]In fact, we can also include time-reversal symmetry, then the Lorentz group is replaced by $O(d)$, and the extension is replaced by $Pin^{\pm}(d)$. In the invertible fermionic topological order of [12], the symmetry extension is $Pin^-(2)$ for $d = 2$ spacetime dimension, which corresponds to time-reversal symmetry that squares to 1 instead of $(-1)^F$.

one-form gauge field, for the $SO(d)$ symmetry satisfies the relation

$$d\rho_1 = w_2\,, \tag{3}$$

where the right hand side denotes the pullback of $w_2 \in H^2(BSO(d), \mathbb{Z}_2)$ to the spacetime manifold $M$ by the gauge field $A : M \to BSO(d)$. Here, remember that a background $A$ for the $SO(d)$ symmetry on the spacetime manifold $M$ can be viewed as a map from $M$ to the classifying space $BSO(d)$. Equation (3) implies that the manifold is equipped with a $Spin(d)$ bundle, with $\rho_1$ plays the role of the spin structure.

In the following we will explore the implications of this relation.

In the presence of background $\rho_1$ for the fermion parity symmetry, the fermionic particle that transforms under the symmetry is attached to a Wilson line $\int \rho_1$ on the world line of the fermionic particle to be gauge invariant. Then the relation (3) implies that the world line has an anomaly: under the background gauge transformation $w_2 \to w_2 + d\lambda_1$, $\rho_1 \to \rho_1 + \lambda_1$. Another way to describe the anomaly is using an auxiliary 1+1d bulk (whose boundary is the worldline) with the effective action

$$\pi \int_\Sigma w_2(TM)\,, \tag{4}$$

which is gauge invariant on closed surface $\Sigma$, and on an open surface it produces the anomalous transformation of $\rho_1$. The boundary of $\Sigma$ carries projective representation of $SO(d)$ described by $w_2$ [20, 21]. As mentioned earlier, a world line of a fermionic particle along a closed loop $l$ needs to be attached to the Wilson line $\oint_l \rho_1$. Given (3), the Wilson line $\oint_l \rho_1$ can be rewritten as a Wilson surface operator $\int_\Sigma w_2(TM)$ with the surface $\Sigma$ such that $\partial\Sigma = l$. For the world line of a local fermionic particle, the value of the Wilson surface $\int_\Sigma w_2(TM)$ attached to it is required to be the same for any choice of the surface $\Sigma$ such that $\partial\Sigma = l$, which equivalently requires $w_2(TM)$ to belong to the trivial cohomology class in $H^2(M, \mathbb{Z}_2)$. Note that this requirement is the same as requiring the spacetime manifold $M$ to be a spin manifold. In fact, choosing a background $\rho_1$ that satisfies the relation (3) with $w_2$ taken to be the $w_2(TM)$ is equivalent to choosing a spin structure on the spacetime manifold $M$. The change of the fermion world line under the transformation $w_2 \to w_2 + d\lambda_1$, $\rho_1 \to \rho_1 + \lambda_1$ is exactly the expected dependence of the fermion world line on the spin structure.[5]

## 2.2 Kitaev's chain as invertible $\mathbb{Z}_2$ one-form gauge theory

Consider $\mathbb{Z}_2$ gauge theory with action

$$\frac{\pi}{2} \int q_{\rho_1}(b)\,, \tag{6}$$

where $b$ is the $\mathbb{Z}_2$ one-form gauge field.

---

[5]We remark that similar but different phenomena is discussed in [22]. An example is $U(1)$ gauge theory whose Wilson line describes the world line of a fermionic particles. This means that the particle satisfies the spin/charge relation *i.e.* particles with odd $U(1)$ charges are fermions, and the dynamical gauge field is more appropriately described by a spin$^c$ connection $a$ that satisfies[6]

$$\int \frac{da}{2\pi} = \frac{1}{2} \int w_2(TM) \mod 1\,. \tag{5}$$

Then the unit charge Wilson line $\oint_\gamma a = \int_\Sigma da$, which describes a fermion, depends on the surface $\Sigma$ that bounds the line by (4): taking two surfaces $\Sigma, \Sigma'$ with the same boundary line produce two operators differed by $da$ integrating over the closed surface $\Sigma \cup \overline{\Sigma'}$, which is (4) by the condition (5).

The quadratic function $q$ satisfies $q_{\rho_1}(x + y) = q_{\rho_1}(x) + q_{\rho_1}(y) + 2x \cup y \mod 4$ for $\mathbb{Z}_2$ one-forms $x, y$; such quadratic functions are labelled by $\mathbb{Z}_2$ one-cochain $\rho_1$, which corresponds to the spin structures. It can also be understood as counting the self intersection of $\mathbb{Z}_2$ valued cycles modulo 4. We summarize some mathematical property of the quadratic function $q$ in Appendix B.

The theory has a unique ground state on any space manifold and it describes an invertible topological order. One way to see this is by stacking the theory with its complex conjugation, whose gauge field denoted by $b'$: the total action is

$$\frac{\pi}{2} \int q_{\rho_1}(b) - \frac{\pi}{2} \int q_{\rho_1}(b') = \frac{\pi}{2} \int q_{\rho_1}(b'') + \pi \int b'' \cup b', \quad b'' = b + b', \qquad (7)$$

where $b'$ acts as a Lagrangian multiplier that forces $b'' = 0$, and thus the total theory is trivial. This shows that the theory (6) is invertible, with the inverse theory given by the complex conjugated theory described by the gauge field $b'$. In particular, the theory is gapped with a unique ground state.

The theory has $\mathbb{Z}_2$ 0-form symmetry generated by the closed loop $\oint b$. Denote the background of the symmetry by $\rho_1$. Due to the quadratic action of the gauge field, the symmetry mixes with the spacetime symmetry to form the extension $Spin(d)$ and it can be identified with the fermion parity symmetry. The background satisfies

$$d\rho_1 = w_2, \qquad (8)$$

where $w_2$ is the $\mathbb{Z}_2$ two-cocycle that represents the second Stiefel-Whitney class of the manifold.

The 0-form symmetry transforms disorder operator of the one-form gauge field $b$. It is a point operator that carries unit holonomy $\oint b = 1$. As the 0-form symmetry is the fermion parity symmetry, the operator is a fermion. Due to the quadratic action of $b$, it also carries gauge charge[7] and needs to attach to the Wilson line $\int b$. We can express the disorder operator as

$$e^{i\phi(p) + i \int_W b}, \qquad (10)$$

where $W$ is a line ending on point $p$ where the disorder operator $\phi$ is inserted. For a closed circle the operator takes value in $\pm 1$ depending on whether the fermion particles obey the anti-periodic or periodic boundary condition, which is the spin structure along the circle as specified by $\oint \rho_1 = 0, 1$. The property of the quadratic function for the intersection form implies that when the fermion particles intersect it produces a $(-1)$, in agreement with the fermion statistics.

The partition function of the theory is given by

$$Z[\rho_1] = \frac{1}{|H_1(M, \mathbb{Z}_2)|^{1/2}} \sum_{b \in H^1(M, \mathbb{Z}_2)} e^{\frac{\pi i}{2} \int q_{\rho_1}(b)}, \qquad (11)$$

---

[7]For instance, we can take the theory on $S^2$ with the disorder operator inserted at a point on $S^2$, and we take $S^2$ to be infinitely elongated along a direction. This means that when expressing $S^2$ as fibration of $S^1$ over an infinite interval $(-\infty, \infty)$ with the disorder operator inserted at $+\infty$, the $S^1$ carries holonomy of the one-form gauge field $b$. In other words, $b = b' + \Omega$ where $\Omega$ is the volume form on $S^1$ and $b'$ is in the perpendicular direction. Then when we take the size of $S^1$ to be small, the system can be described by disorder operator attached to

$$\frac{\pi}{2} \int_{S^2} q_{\rho_1}(b) = \frac{\pi}{2} \int_{S^2} q_{\rho_1}(b' + \Omega) = \pi \int_{\mathbb{R}} (b' + \rho_1), \qquad (9)$$

where we used $q_{\rho_1}(b' + \Omega) = q_{\rho_1}(b') + q_0(\Omega) + 2(\rho_1 + b') \cup \Omega$ for $S^2$, and the integration over $S^1$ picks up to the coefficient of the linear terms for $\Omega$. Thus the disorder operator carries gauge charge and needs to be attached to a Wilson line.

which is known as the Arf-Brown-Kervaire invariant in 1+1d [16]. The partition function is that of the Kitaev's Majorana chain in the non-trivial phase [11,12]. On a spacetime torus the partition function can be expressed as

$$Z[\rho_1] = (-1)^{\text{Arf}(\rho_1)},\tag{12}$$

which equals $(-1)$ for the odd spin structure *i.e.* periodic boundary condition for the fermions along both circles, and $+1$ for the even spin structure.

The theory describes the same low energy physics as a massive Majorana fermion $\psi$, where the fermion parity symmetry maps to the symmetry generated by $\oint b$, and schematically the operators that are transformed under the symmetry map as

$$\text{``}\quad \psi(p) \sim e^{i\phi(p)+i\int_W b}\quad \text{''}.\tag{13}$$

## 3 Invertible exotic loop topological order in 3+1d

In this section we describe the generalization of Kitaev's chain to 3+1d, called the invertible exotic loop topological order (iELTO). We will present the phase as a twisted $\mathbb{Z}_2$ two-form gauge theory that is invertible [25], *i.e.* it is gapped with a unique ground state on any space. The theory has $\mathbb{Z}_2$ one-form symmetry and time-reversal symmetry, which are not a direct product but instead combine into a spacetime two-group symmetry; as a consequence, the one-form symmetry is a part of the spacetime symmetry instead of an internal symmetry. Let us first discuss the property of the symmetry, then we will describe an invertible phase with spacetime two-group symmetry realized by a $\mathbb{Z}_2$ two-form gauge theory. In Section 3.5 we will investigate the boundary properties that characterize the invertible exotic loop topological order. Then we will describe a UV model given by $SO(3)$ gauge theory in Section 3.6.

### 3.1 Global two-group symmetry and its consequences

Let us begin with a discussion of spacetime two-group global symmetry. We give a detailed introduction in Appendix A. The discussion here applies to any system with the spacetime two-group symmetry. In Section 3.2 we will introduce a model (34) for the invertible exotic loop topological order, that describes an invertible phase with spacetime two-group symmetry.

The two-group symmetry extends the $O(4)$ Lorentz symmetry by a $\mathbb{Z}_2$ one-form symmetry

$$1 \to \mathbb{Z}_2 \to \mathbb{G}^{(2)} \to O(4) \to 1.\tag{14}$$

The two-group extension is specified by a class in $H^3(BO(4),\mathbb{Z}_2) = \mathbb{Z}_2[w_1w_2, w_1^3, w_3]$. Notice that we start from the $O(4)$ Lorentz symmetry here because we are interested in systems with time-reversal symmetry. We will discuss the two-group that corresponds to $\omega_3 = w_1w_2 \in H^3(BO(4),\mathbb{Z}_2)$. We denote the two-group by

$$\mathbb{G}^{(2)}[w_1w_2] = \mathbb{Z}_2^{(1)} \times_{w_1w_2} O(4)^{(0)},\tag{15}$$

where $w_1w_2 \in H^3(BO(4),\mathbb{Z}_2)$. The associativity of the $O(4)$ spacetime symmetry transformations is modified,

$$R_{h,h'}R_{hh',h''} = \epsilon^{w_1(h)w_2(h',h'')}R_{h',h''}R_{h,h'h''},\quad h,h',h'' \in O(4),\tag{16}$$

where $\epsilon$ denotes the non-trivial element of the $\mathbb{Z}_2$ one-form symmetry. Here, $R_{h,h'}$ characterizes the difference between the two operations: (1) two consecutive actions of $h'$ and $h$ and (2) a

single action of $hh'$. To be more precise, we should think about the two types of operations on a segment of a loop charged under the $\mathbb{Z}_2^{(1)}$ symmetry. Within the interior of the segment, there is no difference between the two operations. This difference is characterized by the two operators $R_{h,h'}$ and its conjugate with each acting on each end of the segment. The so-defined operator $R_{h,h'}$ characterizes the cohomology class with the 2-group $\mathbb{G}^{(2)}[w_1 w_2]$ via (16).[8] As an example of $(-1)^{w_1 w_2(h,h',h'')} = -1$, if we take $h',h'' \in O(4)$ such that $\pi_{h'}\pi_{h''} = -\pi_{h'h''}$ for a defect carrying spinor projective representation $\pi$ (it will not be a genuine particle since it is not a linear representation),[9] then $w_2(h',h'') = 1 \in \{0,1\} = \mathbb{Z}_2$. Then for $h = \mathcal{T}$ the time-reversing transform in $O(4)$ , $w_1(h) = 1$, and $(-1)^{w_1 w_2(h,h',h'')} = -1$.

The background gauge fields for the two-group symmetry (15) obey the following relation

$$d\rho_2 = w_1 w_2,\tag{17}$$

where $\rho_2$ is the background gauge field for the $\mathbb{Z}_2$ one-form symmetry, and $w_1 w_2$ above are the pullback of $w_1 w_2 \in H^3(BO(4), \mathbb{Z}_2)$ by the $O(4)$ background gauge field $A : M \to BO(4)$ to the spacetime manifold $M$. In the following we will investigate several implications of the above relation.

### 3.1.1 Time-reversal symmetry

Let us discuss in more detail the time-reversal symmetry of the theory. Being part of the spacetime two-group symmetry, the time reversal symmetry will transform $\rho_2$, the background gauge field of the $\mathbb{Z}_2$ one-form symmetry. Let us see how it works. From the relation between background gauge fields

$$d\rho_2 = w_1 w_2,\tag{18}$$

the time-reversal symmetry transformation reverses the local orientation on the spacetime by

$$w_1 \to w_1 + d\lambda_0,\tag{19}$$

where $\lambda_0 = 0, 1 \in \mathbb{Z}_2$, and a nonzero value reverses the orientation. Here, $w_1$ describes the orientation bundle[10]. It follows that

$$\rho_2 \to \rho_2 + \lambda_0 w_2,\tag{20}$$

where we have used $dw_2 = 0$. Plugging in $\lambda_0 = 1$, it means that $\rho_2$ also shifts by

$$\mathcal{T}: \quad \rho_2 \to \rho_2 + w_2.\tag{21}$$

Therefore, we conclude that on an orientable manifold ($w_1 = 0$), the time reversal symmetry in the two-group is the ordinary time reversal symmetry together with a shift of the background gauge field $\rho_2$ for the one-form symmetry part of the two-group symmetry by $w_2$. This additional shift of $\rho_2$ is first studied in [26] as the fractionalization map.

---

[8]For a similar discussion in 2+1d SPT phase, see *e.g.* [20]; here, the boundary of the SPT is the worldsheet of the string charged under the one-form symmetry, see Section 3.1.2 for more detail.

[9]$SO(4) = (SU(2) \times SU(2))/\mathbb{Z}_2$, where the two $SU(2)$s describe the six rotations on the six planes in $\mathbb{R}^4$. The representations are labelled by $(j_1, j_2)$ for $SU(2)$ spin $j$. Projective spinor representations are those with $j_1 - j_2 \in \mathbb{Z} + 1/2$. An example is $j_1 = 1/2, j_2 = 0$. For instance, we can take $h, h''$ to be the $\pi$ rotation on the $x, y$-plane in $(t, x, y, z)$ coordinate, then they compose to a $2\pi$ rotation, which acts on the projective spinor representation as $-1$.

[10]The manifold is unorientable if and only if $w_1$ is non-trivial, where a one-cycle is unorientable if and only if $w_1$ has non-trivial holonomy on it. One can roughly think of it as the "gauge field for the time-reversal symmetry".

Let us investigate the property of any loop $W$ that is charged under the one-form symmetry part in the two-group. In the presence of background gauge field $\rho_2$ for the one-form symmetry, under a background gauge transformation $\rho_2 \to \rho_2 + d\lambda_1'$ the loop $W$ transforms as (with parameter given by one-form $\lambda_1'$)

$$W \to W e^{i \oint \lambda_1'}. \tag{22}$$

Thus $W$ is not invariant, and it needs to be attached to the boundary of a Wilson surface $\int \rho_2$ to form a one-form gauge invariant configuration. That is to say the world history of the loop needs to be stacked with the Wilson surface $\oint \rho_2$. Since the world history of the loop that transforms under the one-form symmetry is stacked with $\int \rho_2$, the background gauge field for the one-form symmetry, to be invariant under the one-form transform, the transformation (21) implies that the world history surface is stacked with additional $\int w_2$ under the time-reversal symmetry. The stacking of an additional $\int w_2$ to the world history surface of the loop can be rephrased as the stacking of an additional world line that carries the spinor representation of the Lorentz group to the boundary of the world history surface, i.e. the loop itself.

We remark that the loop excitation transforms as a representation of the two-group symmetry [27], which includes the time-reversal symmetry. It is also a projective 2-representation [28] of the Lorentz group (which is a category equipped with a non-associative symmetry action), similar to the property that fermion particle transforms as a projective representation of the Lorentz group.

In the case without time-reversal symmetry, $w_1 = 0$, the equation (17) no longer describes a non-trivial extension of the spacetime rotation group $SO(4)$ by the $\mathbb{Z}_2$ one-form symmetry. The two-group reduces to the product of a 0-form symmetry and one-form symmetry that are unrelated. In this case the two-group SPT phases become the previously known SPT phases with $SO(4)$ spacetime rotation symmetry and a factorized $\mathbb{Z}_2$ one-form symmetry. In other words, the SPT phase protected by the non-trivial two-group symmetry $\mathbb{G}^{(2)}[w_1 w_2] = \mathbb{Z}_2^{(1)} \times_{w_1 w_2} O(4)^{(0)}$ reduce to the SPT phases protected by $\mathbb{G}^{(2)} = \mathbb{Z}_2^{(1)} \times SO(4)^{(0)}$ if the time-reversal symmetry is broken explicitly.

In Section 3.4.1 we will show the partition function is invariant under the modified time-reversal symmetry in the two-group spacetime symmetry.

We remark that since performing a time-reversal transformation on an open worldsheet of the exotic loop produces additional spinor projective representation of the Lorentz symmetry on the boundary of the worldsheet, which is a loop in 3+1d. The spinor projective representation makes the correlation function depend on the framing of the worldsheet. For instance, performing a $2\pi$ rotation on the normal vectors along the boundary of the worldsheet (after the time-reversal transformation) changes the correlation function by a sign. Thus, the correlation function depends on the normal vectors on the worldsheet.[11] Similar framing dependence of the correlation function of open surfaces on $S^4$ was also discussed in [29].

### 3.1.2 Lorentz symmetry acts anomalously on exotic loop

The theory describes an invertible phase protected by the two-group symmetry. Recall that in an ordinary 0-form $G$ SPT phase in $2+1$ dimensions, on its boundary, there are excitations on which the symmetry acts anomalously– with an associator $\omega_3 \in H^3(BG, U(1))$ [20]. Here, in the invertible phase protected by two-group symmetry (15), there is a similar but more peculiar phenomenon. The $O(4)$ symmetry acts on the loop carrying one-form symmetry charge anomalously, with the associator $(-1)^{w_1 w_2} \in H^3(BO(4), U(1))$. This can be understood from

---

[11]More generally, we can perform the time-reversal transformation on a region of the worldsheet. Then the boundary of the region on the worldsheet will carry additional spinor projective representation according to Section 3.1.1. As the region is arbitrary, the correlation function depends on the normal vectors on the entire worldsheet, not just the normal vectors on the boundary.

the relation between the symmetry actions of the $O(4)$ symmetry and the $\mathbb{Z}_2$ one-form symmetry, given by (16). Since the loop is transformed by a sign under the $\mathbb{Z}_2$ one-form symmetry, the $O(4)$ symmetry action on the loop is not associative, but only up to a sign:

$$R_{h,h'}R_{hh',h''} = -R_{h',h''}R_{h,h'h''}, \qquad \text{if } w_1 w_2(h, h', h'') = 1. \tag{23}$$

Alternatively, the anomalous symmetry action on the loop can be understood from the transformation of the loop in the presence of background gauge fields, which obey the relation (17). The symmetry action (16) implies that when the 0-form symmetry defect undergoes an F-move, which can be implemented by a gauge transformation, there is an additional one-form symmetry defect. We turn on the background gauge fields $A$ for the Lorentz symmetry $O(4)$ and the $\rho_2$ for $\mathbb{Z}_2$ one-form symmetry. Consider a gauge transformation for $O(4)$,

$$w_1 \to w_1 + d\lambda_0, \qquad w_2 \to w_2 + d\lambda_1, \tag{24}$$

with $\mathbb{Z}_2$ scalar $\lambda_0$ and $\mathbb{Z}_2$ one-form $\lambda_1$. Then the relation (17) implies that the gauge field $\rho_2$ also undergoes a gauge transformation,

$$\rho_2 \to \rho_2 + \lambda_0 w_2 + w_1 \lambda_1 + \lambda_0 d\lambda_1. \tag{25}$$

In other words, the gauge transformations for $A$ and $\rho_2$ are not independent.

In the presence of background gauge field $\rho_2$ for the one-form symmetry, any loops $W$ that transforms under the one-form symmetry is not by itself invariant under the one-from transform, and needs to be stacked with the Wilson surface $\int \rho_2$ to form a one-form gauge invariant configuration. On the other hand, $\int \rho_2$ is not invariant under the $O(4)$ transformations of backgrounds $w_1, w_2$, but transforms as in (25). Thus, the world history surface of the loop, which is stacked with the Wilson surface $\int \rho_2$, is not invariant under the $O(4)$ transformations for backgrounds $w_1, w_2$, i.e. the $O(4)$ symmetry acts anomalously on the loop.

The anomaly can also be described by an auxiliary three-dimensional bulk $\mathcal{V}$ that bounds the loop world history (when it is a closed surface), with the effective action

$$\pi \int_{\mathcal{V}} w_1 w_2(TM)|_{\mathcal{V}}, \tag{26}$$

where $w_1 w_2(TM)|_{\mathcal{V}}$ denotes the restriction to $\mathcal{V}$. The effective action is gauge invariant on closed manifolds, and on manifolds with boundaries the transformation (24) produces the boundary term $\pi \int (\lambda_0 w_2 + w_1 \lambda_1 + \lambda_0 d\lambda_1)$ as in (25).

Similar to the discussion in Section 2.1, when we want to construct invertible topological phases using the exotic loops as the elementary degrees of freedom, we will need to make sure that the world history surface of the exotic loop does not depends on the choice of the 3d manifold $\mathcal{V}$. This requirement means that we will only focus on the spacetime manifolds $M$ such that $w_1 w_2(TM)$ belongs to trivial cohomology class in $H^3(M, \mathbb{Z}_2)$, namely spacetime manifolds $M$ that admits a $v_3$ Wu structure. A $v_3$ Wu structure, or Wu structure of degree 3, on a spacetime manifold $M$ is specified by a background $\rho_2$ that satisfies the condition (18) with the $w_1 w_2$ term taken to be $w_1 w_2(TM)$ which is known as the third Wu class $v_3(TM) = w_1 w_2(TM)$ of the spacetime manifold. The invertible topological phase that is constructed from the exotic loop, namely that is protected by the two-group symmetry $\mathbb{G}^{(2)}[w_1 w_2]$, has a partition function that depends on the background $\rho_2$, i.e. the $v_3$ Wu structure.

### 3.1.3 Consequence of breaking $\mathbb{Z}_2$ one-form symmetry

The $\mathbb{Z}_2$ one-form symmetry cannot be explicitly broken without also breaking the time-reversal symmetry, or introducing local fermionic particles to the system. This can be understood using

the relation between background fields

$$d\rho_2 = w_1 w_2 \,, \tag{27}$$

which implies that if we were to break the $\mathbb{Z}_2$ one-form symmetry, as there is no longer a $\mathbb{Z}_2$ one-form symmetry, the system cannot couple to a non-trivial background in a gauge invariant way, and we are forced to have $\rho_2 = 0$. Then it is not consistent with a nonzero right hand side unless $w_1 = 0$ or $w_2 = 0$. The latter condition changes the spacetime two-group symmetry: $O(4)$ symmetry is replaced by $SO(4)$ or $Pin^+(4)$ in the two cases, respectively. The former case $w_1 = 0$ would restrict the gauge field to be that of $SO(4)$ symmetry, and the system would no longer have time-reversal symmetry. In a system with time-reversal symmetry, one can consider the twisted boundary condition under time-reversal symmetry along one direction. In this system, $w_1$ is non-trivial. Thus there is no time-reversal symmetric system where $w_1$ is always trivial. The other case $w_2 = 0$ restricts the gauge field to be that of $Pin^+(4)$ symmetry, which implies that the system has local fermionic particles with the property the time-reversal element of $Pin^+(4)$ squares to $(-1)^F$ where $(-1)^F$ is the fermion parity operator of the local fermion [16].

### 3.1.4 Symmetry in the renormalization group flow

It often occurs that the symmetry in a microscopic system, such as lattice model, differs from the symmetry in the low energy physics. Suppose the two-group symmetry is approximate, and it only emerges in the low energy physics. We will derive a constraint on the symmetry in the microscopic model for the iELTO phase, similar to the discussion in [21, 30, 31].

Suppose the microscopic system has 0-form symmetry $G_{\mathrm{UV}}^{(0)}$ and one-form symmetry $G_{\mathrm{UV}}^{(1)}$. They map to the symmetry at low energies by homomorphisms $f_0, f_1$,

$$f_0: \quad G_{\mathrm{UV}}^{(0)} \to G_{\mathrm{IR}}^{(0)} = O(4), \qquad f_1: \quad G_{\mathrm{UV}}^{(1)} \to G_{\mathrm{IR}}^{(1)} = \mathbb{Z}_2 \,. \tag{28}$$

In general, the UV 0-form symmetry and one-form symmetry can also be described by a two-group, with potentially trivial Postnikov class $\omega_3^{\mathrm{UV}} \in H^3(BG_{\mathrm{UV}}^{(0)}, G_{\mathrm{UV}}^{(1)})$ that describes the $G_{\mathrm{UV}}^{(1)}$-valued associator of the 0-form symmetry $G_{\mathrm{UV}}^{(0)}$. If we turn on background gauge fields $B_2^{\mathrm{UV}}, A^{\mathrm{UV}}$ for the UV one-form and 0-form symmetries, they satisfy

$$dB_2^{\mathrm{UV}} = (A^{\mathrm{UV}})^* \omega_3^{\mathrm{UV}} \,, \tag{29}$$

where star denotes the pullback by the gauge field. Then for the relations (29) and (17) to be compatible, $\omega_3^{\mathrm{UV}}$ is constrained to satisfy

$$f_1 \omega_3^{\mathrm{UV}} = f_0^* \omega_3^{\mathrm{IR}}, \quad \omega_3^{\mathrm{IR}} = w_1 w_2 \,. \tag{30}$$

For instance, if the UV symmetry is a trivial two-group with $\omega_3^{\mathrm{UV}} = 0$ (for instance, when there is no one-form symmetry), then $f_0$ must satisfy $f_0^* \omega_3^{\mathrm{IR}} = 0$ and it is not the identity map: the time-reversal symmetry is broken, namely, the UV Lorentz symmetry is only $SO(4)$, or the $O(4)$ symmetry is extended to be $Pin^+(4)$.

### 3.1.5 Extending the two-group symmetry by fermion parity

Although the iELTO phase does not have local fermion particles and $\mathbb{Z}_2$ fermion parity symmetry, let us contemplate the situation when we stack the iELTO phase with a fermionic SPT phase with time-reversal symmetry that satisfies $\mathcal{T}'^2 = (-1)^F$ and leaves $\rho$ invariant, where we used different notation $\mathcal{T}'$ to distinguish this symmetry from the time-reversal symmetry

$\mathcal{T}$ in (21).[12] We will show that in such case the spacetime two-group symmetry is extended by the fermion parity symmetry to be the direct product of internal $\mathbb{Z}_2^{(1)}$ one-form symmetry and $Pin^+(4)$ Lorentz symmetry in which the time-reversal satisfies $\mathcal{T}'^2 = (-1)^F$:

$$1 \to \mathbb{Z}_2^F \to \mathbb{Z}_2^{(1)} \times Pin^+(4) \to \mathbb{G}^{(2)}[w_1 w_2] \to 1. \tag{31}$$

Conversely, the direct product of an internal $\mathbb{Z}_2$ one-form symmetry and $Pin^+(4)$ symmetry can be expressed as the extension of the spacetime two-group symmetry with additional fermion parity symmetry.

The extension can be understood using their background gauge fields as follows. In the presence of fermion parity symmetry, we need to consider spin (or pin$^+$ manifold if unorientable) instead of general manifolds, where $[w_2] = 0 \in H^2(M, \mathbb{Z}_2)$ is exact *i.e.* $w_2 = dz$ for a $\mathbb{Z}_2$ one-cochain $z$ that represents the spin (or pin$^+$) structure. [13] Here, $[w_2]$ denotes the cohomology class $w_2$ represents in $H^2(M, \mathbb{Z}_2)$. The two-group background satisfies $d\rho_2 = w_1 w_2$. Then the field redefinition $\tilde{\rho}_2 = \rho_2 - w_1 z$ with $dz = w_2$ being the spin structure, satisfies $d\tilde{\rho}_2 = 0$ and it is the background for the $\mathbb{Z}_2$ one-form symmetry that is independent of the spacetime symmetry $Pin^+(4)$.

The symmetry extension $\mathbb{Z}_2^{(1)} \times Pin^+(4)$ contains the following two time-reversal symmetries:

1. $\mathcal{T}'^2 = (-1)^F$, independent of the one-form symmetry. This is the ordinary time-reversal symmetry in $Pin^+(4)$.

2. Modified time-reversal symmetry $\mathcal{T}$, independent of the fermion parity symmetry generated by $(-1)^F$. This is the time-reversal symmetry in the spacetime two-group $\mathbb{G}^{(2)}[w_1 w_2]$. It changes the spin of the loop that transforms under the one-form symmetry.

The second time-reversal symmetry was discussed in Section 3.1.1. Under the time reversal transformation,[14]

$$\int_\Sigma \rho_2' = \int_\Sigma \rho_2 + \oint_\gamma z, \quad \gamma = \partial \Sigma. \tag{32}$$

Thus after the time-reversal transformation, the exotic loop that is charged under the one-form symmetry is attached to the Wilson line of the spin structure,

$$\mathcal{W}_\gamma e^{\pi i \int_\Sigma \rho_2'} = \left( \mathcal{W}_\gamma e^{\pi i \oint_\gamma z} \right) e^{\pi i \int_\Sigma \rho_2}. \tag{33}$$

The spin structure is the gauge field for $(-1)^F$ symmetry which transforms spinor representations. Thus $\mathcal{W}_\gamma$ carries additional spinor representation after the time-reversal transformation, just as in the discussion of Section 3.1.1.

## 3.2 Invertible phase as $\mathbb{Z}_2$ two-form gauge theory

The model can be described by a $\mathbb{Z}_2$ two-form gauge theory with gauge field $b$. It has the following action,

$$S[\rho_2] = \frac{\pi}{2} \int q_{\rho_2}(b), \tag{34}$$

---

[12]For instance, if we consider theories on a spin (or pin$^+$ if unorientable) manifold, the Lorentz group $SO(4)$ (or $O(4)$) is extended by $\mathbb{Z}_2^F$ fermion parity symmetry (that transforms the spinors by $(-1)$) to be $Spin(4)$ ($Pin^+(4)$). The manifold is equipped with a gravitational Wilson line of the spin connection in the spinor representation of $Spin(4)$ ($Pin^+(4)$), and it is the world line of a decoupled neutral massive fermion particle.

[13]It is a $H^1(M, \mathbb{Z}_2)$ torsor.

[14]For instance, $\Sigma$ can be a higher-genus Riemann surface with boundary $\gamma$.

where $b$ is a $\mathbb{Z}_2$ two-form gauge field $b$. This gauge theory is defined through a quadratic form $q_{\rho_2}(b)$. We summarize some mathematical properties of the quadratic function $q$ in Appendix B. The theory is invertible, *i.e.* gapped with a unique ground state, by a similar argument as in (7) using the property of the quadratic function $q$. $\rho_2$ is the background gauge field for $\mathbb{Z}_2$ one-form symmetry, and the subscript 2 indicates it is a 2-form gauge field. To simplify notation, in this section we will sometimes drop the subscript and write it as $\rho$. As a property of the quadratic form in four dimensional spacetime, $\rho_2$ is specified by

$$d\rho_2 = v_3 = w_1 w_2 \in H^3(BO(4), \mathbb{Z}_2), \tag{35}$$

where $v_3$ is the third Wu class. As we discussed in Section 3.1, this means that the theory has a spacetime two-group symmetry. This symmetry implies that the loops, which are charged under the one-form symmetry part of the two-group, have exotic properties as discussed in detail in Section 3.1. Equation (35) implies that the manifold is equipped with a two-group bundle, with $\rho_2$ plays the role of the $v_3$ Wu structure.

The disorder operator, denoted by $W$, that carries unit flux of the two-form gauge field $b$, transforms under the one-form symmetry. The disorder operator $W$ is defined on a closed loop. Due to the quadratic action, it also carries one-form gauge charge[15] and needs to attach to the Wilson surface $e^{\pi i \int b}$. Thus we can express the disorder operator as the open Wilson surface $We^{\pi i \int b}$ (in the presence of background $\rho_2$ for the one-form symmetry, it needs to be attached with the Wilson surface $e^{\pi i \int \rho_2}$ to be invariant under the one-form gauge transformation of $\rho_2$). The world history of the loop is described by a closed surface operator $e^{\pi i \oint b}$, and it is also the generator of the $\mathbb{Z}_2$ one-form symmetry.

The action counts the mod-4 self-intersection number for the loops with world history dual to $b$, denoted by $\tilde{b}$:

$$e^{\frac{\pi i}{2} \left( \#\text{self-intersection of } \tilde{b} \right)} = e^{i \frac{\pi}{2} \int q_{\rho_2}(b)}. \tag{37}$$

The self-intersection number determined by the quadratic function $q$ needs to be defined by a regularization and it can be an integer in the above expression.[16] Different regularizations correspond to different $\rho$. They all give the same mutual intersection number, namely $q_{\rho_2}(b+b') = q_{\rho_2}(b) + q_{\rho_2}(b') + 2b \cup b'$. In particular, when two loops intersect the action produce a $\pi$ phase. This is a generalization of the $\pi$ phase appearing when two fermion worldlines intersect in the $1+1d$ $\mathbb{Z}_2$ theory (6) describing the Kitaev's chain invertible fermion topological order. As the intersection between a surface and an open surface with boundary equals the linking between the surface and the boundary of the open surface, the $\pi$ phase intersection is in agreement with the property that loop, which lives on the boundary its world history, transforms non-trivially under the one-form symmetry generated by the world history surface

---

[15]For instance, we can take the theory on $S^3 \times S^1$ with the disorder loop $W$ wraps $S^1$ and inserted at a point on $S^3$, and we take $S^3$ to be infinitely elongated along a direction. This means that when expressing $S^3$ as fibration of $S^2$ over an infinite interval $(-\infty, \infty)$ with the disorder operator inserted at $+\infty$, the $S^2$ carries flux of the two-form gauge field $b$. In other words, $b = b' + \Omega$ where $\Omega$ is the volume form on $S^2$ with $\oint \Omega = 1$, and $b'$ is the two-form gauge field in the perpendicular directions. Then when we take the squashing limit with vanishing size of $S^2$, the system can be described by disorder operator attached to

$$\frac{\pi}{2} \int_{S^3 \times S^1} q_{\rho_2}(b) = \frac{\pi}{2} \int_{S^3 \times S^1} q_{\rho_2}(b' + \Omega) = \pi \int_{\mathbb{R} \times S^1} (b' + \rho_2), \tag{36}$$

where we used $q_{\rho_2}(b' + \Omega) = q_\rho(b') + q_0(\Omega) + 2(b' + \rho_2) \cup \Omega$ for $S^3 \times S^1$, and the integral over $S^2$ is nonzero only when integrated over linear terms of $\Omega$. Thus the disorder operator carries one-form gauge charge and needs to be attached to a Wilson surface.

[16]On a manifold with a spin structure, the quadratic function $q$ is an even integer. Odd $q$ (*i.e.* odd self-intersection number in (37)) can appear on a manifold without a spin structure. For example, on the complex projective plane $\mathbb{CP}^2$, the complex projective line $\mathbb{CP}^1 \subset \mathbb{CP}^2$ has the minimal self-intersection number $\#(\mathbb{CP}^1, \mathbb{CP}^1) = 1$ [32].

$e^{\pi i \oint b}$. We compute the correlation function of the worldsheet of the exotic loop in Section 3.4.2.

## 3.3  $\mathbb{Z}_8$ classification

Let us show that the invertible phase has $\mathbb{Z}_8$ classification. Consider four copies of the theory described by four independent $\mathbb{Z}_2$ two-form gauge fields $b^i$, $i = 1, 2, 3, 4$. The action is given by

$$\sum_{i=1}^{4} \frac{\pi}{2} q_{\rho_2}(b^i). \tag{38}$$

By a change of variables

$$b'^1 = b^1, \quad b'^2 = b^2 + b^3 + b^4, \quad b'^3 = b^3 + b^2, \quad b'^4 = b^4 + b^2, \tag{39}$$

the theory is equivalent to

$$\left( \frac{\pi}{2} q_{\rho_2}(b'^1) - \frac{\pi}{2} q_{\rho_2}(b'^2) \right) + \pi (b'^3)^2 + \pi (b'^4)^2 + \pi b'^3 b'^4. \tag{40}$$

The first term describes the phase in the trivial class, while the remaining terms can be simplified using the Wu formula $(b'^4)^2 = b'^4 (w_2 + w_1^2)$, and integrating out the Lagrangian multiplier $b'^4$ results in

$$\pi w_2^2 + \pi w_1^4. \tag{41}$$

It describes the bosonic SPT phase with time-reversal symmetry, whose partition function is $-1$ on both $\mathbb{RP}^4$ and $\mathbb{CP}^2$. In fact, the theory of $b'^3, b'^4$ describes the low energy theory of the three-fermion Walker Wang model, which has boundary state given by the three-fermion theory, where the fermions are Kramers singlet [18, 23, 33, 34].[17] Since two copies of the bosonic SPT phase with time-reversal symmetry is trivial, we conclude that the classification of the new phases is $\mathbb{Z}_8$, with the fourth class differing from the trivial class by the above bosonic time-reversal symmetric SPT phase.[18]

If we take instead two copies of the iELTO, described by

$$\frac{\pi}{2} q_{\rho_2}(b^1) + \frac{\pi}{2} q_{\rho_2}(b^2) = \frac{\pi}{2} q_{\rho_2}(b'^1) + \pi (b'^2)^2 + \pi b'^1 b'^2, \tag{42}$$

where $b'^1 = b^1 + b^2$, $b'^2 = b^2$. Then using the Wu formula $(b'^2)^2 = (w_2 + w_1^2) b'^2$ and integrating out $b'^2$ we find $b'^1 = w_2 + w_1^2$, and thus two copies of the iELTO phase has the effective action

$$\frac{\pi}{2} q_{\rho_2}(w_2 + w_1^2). \tag{43}$$

Since the quadratic function has order 4, this is also consistent with the $\mathbb{Z}_8$ classification.

---

[17]One way to understand this is to start from an ordinary 2+1d $\mathbb{Z}_2$ gauge theory, which has $\mathbb{Z}_2 \times \mathbb{Z}_2$ one-form symmetry that acts on the bosonic electric and magnetic line $e, m$, with anomaly described by the SPT phase in 3+1d $\pi \int B^e \cup B^m$ where $B^e, B^m$ are the background $\mathbb{Z}_2$ two-forms for the one-form symmetry. Then coupling the theory to $B^e = B^m = w_2 + w_1^2$ turns the electric and magnetic particles into Kramers singlet fermions, with the anomaly described by the SPT phase $\pi \int (w_2 + w_1^2)^2 = \pi \int w_2^2 + w_1^4$.

[18]We remark that a similar reasoning implies that the invertible femrionic topological order of [12] in 1+1d with fermion parity and time reversal symmetry (that squares to 1 instead of $(-1)^F$) has $\mathbb{Z}_8$ classification, in agreement with the result in [12, 16].

## 3.4 Partition functions

In the following we will study the partition function of the model (34) describing the invertible exotic loop topological order. The theory has partition function

$$Z^{\text{iELTO}}[\rho_2] = \frac{1}{|H^2(M, \mathbb{Z}_2)|^{1/2}} \sum_{b \in H^2(M, \mathbb{Z}_2)} e^{\frac{\pi i}{2} \int q_{\rho_2}(b_2)}. \tag{44}$$

The partition function on general manifolds is an 8th root of unity, which leads to $\mathbb{Z}_8$ classification of the invertible phase. In the special case when the manifold is orientable, the partition function can be expressed as [13, 35, 36]

$$Z^{\text{iELTO}} = \exp\left(\frac{i}{96\pi} \int \text{Tr} \, R \wedge R - \frac{2\pi i}{4} \int \mathcal{P}(\rho_2)\right), \tag{45}$$

where $\mathcal{P}(\rho_2)$ is the quadratic function $q_{\rho_2'}(B)$ with $\rho_2' = 0, B = \rho_2$, and it coincides with the Pontryagin square operation of $\rho_2$. Note that on an oriented spacetime manifold $M$, it is always valid to take the background $\rho_2' = 0$ in $q_{\rho_2'}(B)$, since $w_1(TM) = 0$. The second term has order 4, while the first term has order 8 on general orientable manifolds.[19] The first term can also be written in terms of the signature $\sigma(M)$ of the manifold $M$:

$$\exp\left(i\pi \frac{\sigma(M)}{4}\right), \tag{46}$$

where $\sigma(M) = \frac{1}{24\pi^2} \int_{\mathcal{M}} \text{Tr} \, R \wedge R$.

### 3.4.1 Time-reversal symmetry of partition function

We can also examine the time-reversal symmetry of the following partition function on general orientable manifolds

$$Z^{\text{iELTO}}[\rho_2] = Z^{\text{iELTO}}[0] e^{-\frac{2\pi i}{4} \int q_0(\rho_2)}, \tag{47}$$

where $q_0$ is the Pontryagin square operation, and

$$Z^{\text{iELTO}}[0] = \sum_b e^{\frac{\pi i}{2} \int q_0(b)} = e^{i\pi \frac{\sigma(M)}{4}}. \tag{48}$$

The time-reversal transformation flips the sign of the action and also transforms $\rho_2$ as in (21) on an orientable manifold. This gives

$$
\begin{aligned}
Z^{\text{iELTO}}[\rho_2]' &= Z^{\text{iELTO}}[\rho_2 + w_2]^* = Z^{\text{iELTO}}[0]^* e^{\frac{2\pi i}{4} \int q_0(\rho_2 + w_2)} \\
&= Z^{\text{iELTO}}[0]^* e^{\frac{2\pi i}{4} \int q_0(w_2)} e^{-\frac{2\pi i}{4} \int q_0(\rho_2)}.
\end{aligned} \tag{49}
$$

In the last expression[20], the first two terms combine into[21]

$$Z^{\text{iELTO}}[0]^* e^{\frac{\pi i}{2} \int q_0(w_2)} = Z^{\text{iELTO}}[0]. \tag{51}$$

Thus the theory is time-reversal invariant, namely

$$Z^{\text{iELTO}}[\rho_2]' = Z^{\text{iELTO}}[\rho_2]. \tag{52}$$

---

[19] On the other hand, spin manifolds cannot detect the first term which becomes trivial, and the second term has order 2. In fact, on spin manifolds the partition function is real and thus invariant under the ordinary time-reversal symmetry, see Section 3.1.5 for details.

[20] To derive the last expression, we have used that $q_0(\rho_2 + w_2) = q_0(w_2) + q_0(\rho_2) + 2\rho_2 \cup w_2$ and $e^{i\frac{\pi}{2} \int (q_0(\rho_2) + 2\rho_2 \cup w_2)} = e^{i\frac{\pi}{2} \int (q_0(\rho_2) + 2\rho_2 \cup \rho_2)} = e^{-i\frac{\pi}{2} \int q_0(\rho_2)}$.

[21] The equality follows from

$$Z^{\text{iELTO}}[0]^* e^{\frac{2\pi i}{4} \int q_0(w_2)} = \sum_b e^{-\frac{\pi i}{2} \int q_0(b) + \pi i \int b^2} = \sum_b e^{\frac{\pi i}{2} \int q_0(b)} = Z^{\text{iELTO}}[0]. \tag{50}$$

Also note that on a spin manifold, $e^{i\frac{\pi}{2} \int \rho_0(w_2)} = 1$, and $Z^{\text{iELTO}}[0] = 1$.

### 3.4.2 Correlation function

Let us compute the correlation function of a world-history surface $\oint_\Sigma b$ on a general manifold. The correlation function on $S^4$ spacetime was also discussed in Section 7 of [29] in the absence of background $\rho_2$ and without imposing the two-group symmetry. The correlation function is normalized by the partition function without insertions (44). For simplicity, let us begin by computing the correlation function on orientable manifold with vanishing background $\rho_2$. Consider the correlation function of world-history surface of the loop, $W(\Sigma) = \exp\left(\pi i \oint_\Sigma b\right) = \exp\left(\pi i \int b \cup \delta_2(\Sigma)^\perp\right)$ with delta function two-form $\delta_2(\Sigma)^\perp$ that restricts the integral to $\Sigma$:

$$
\begin{aligned}
\langle W(\Sigma) \rangle &= \frac{1}{|H^2(M, \mathbb{Z}_2)|Z[0]} \sum_{b \in H^2(M, \mathbb{Z}_2)} e^{i\pi \int b \cup \delta_2(\Sigma)^\perp} e^{i\frac{\pi}{2} \int_M q_0(b)} \\
&= \frac{1}{|H^2(M, \mathbb{Z}_2)|Z[0]} \sum_{b' \in H^2(M, \mathbb{Z}_2)} e^{i\frac{\pi}{2} \int_M q(b')} e^{-i\frac{\pi}{2} \int_M q_0(\delta_2(\Sigma)^\perp)},
\end{aligned}
\tag{53}
$$

where $b' = b + \delta_2(\Sigma)^\perp$, and we have used
$q_0(b') - q_0(\delta_2(\Sigma)^\perp) = q_0(b + \delta_2(\Sigma)^\perp) - q_0(\delta_2(\Sigma)^\perp) = q_0(b) + 2b \cup \delta_2(\Sigma)^\perp$. Therefore, the correlation function equals the self-intersection number of $\Sigma$ in (37) for $\rho_2 = 0$:

$$
\langle W(\Sigma) \rangle = e^{-i\frac{\pi}{2} \int_M q_0(\delta_2(\Sigma)^\perp)}.
\tag{54}
$$

In the presence of non-trivial background $\rho_2$, following the similar derivation, the correlation function becomes

$$
\langle W(\Sigma) \rangle = e^{-i\frac{\pi}{2} \int_M q_{\rho_2}(\delta_2(\Sigma)^\perp)}.
\tag{55}
$$

We remark that the correlation function depends on the background $\rho_2$. If we change the background by $\rho_2 \to \rho_2 + z$ for some $z \in H^2(M, \mathbb{Z}_2)$, the correlation function for $e^{i\pi \oint b}$ is changed by replacing $e^{i\pi \oint b}$ with $e^{i\pi \oint b+z}$:[22]

$$
\begin{aligned}
\langle e^{\pi i \oint b} \rangle_{\rho_2 + z} &= \frac{1}{|H^2(M, \mathbb{Z}_2)|^{1/2} Z[\rho_2 + z]} \sum_b e^{\frac{\pi i}{2} \int q_{\rho_2 + z}(b)} e^{\pi i \oint b} \\
&= \frac{1}{|H^2(M, \mathbb{Z}_2)|^{1/2} Z[\rho_2]} \sum_b e^{\frac{\pi i}{2} \int q_{\rho_2}(b)} e^{\pi i \oint b+z} = \langle e^{\pi i \oint b+z} \rangle_{\rho_2},
\end{aligned}
\tag{57}
$$

where in the second equality we redefine $b \to b+z$, and we normalize the correlation function by the partition function $Z[\rho]$ without insertions. Therefore, the correlation function of the world-history of the loop $W(\Sigma) = e^{i\pi \oint_\Sigma b}$ depends on the background gauge field $\rho_2$: changing $\rho_2$ can change the correlation function by $\pm 1$.

## 3.5 Boundary properties of iELTO

The iELTO is protected by both the one-form symmetry and the zero-form spacetime symmetry. On its boundary, both the one-form symmetry and the time-reversal symmetry are realized anomalously. In particular, as we will show, the boundary has a particle-like excitation with anti-semionic self-statistics, and chiral central charge.

These boundary property of iELTO can be derived from the partition function that depends on the background fields ("symmetry twists"). The partition function is well-defined on a

---

[22]We used the identity

$$
q_{\rho_2 + z}(b) = q_{\rho_2}(b) + 2bz = q_{\rho_2}(b+z) - q_{\rho_2}(z).
\tag{56}
$$

closed manifold, while on a manifold with boundary the partition function is not invariant but must be accompanied by some boundary degrees of freedom, which characterizes the bulk phase by the bulk-boundary correspondence.

### 3.5.1 Anomalous one-form symmetry on the boundary

The partition of iELTO phase, $Z^{\text{iELTO}}[\rho_2]$, depends on the background $\rho_2$ for the $\mathbb{Z}_2$ one-form symmetry part of the two-group. Let us fix some $\rho_2 = \rho_\star$, then a general background $\rho_2$ for the one-form symmetry can be written as $\rho_2 = \rho_\star + z$ with $\delta z = 0$ that parametrizes the background $\rho_2$. The partition function depends on the background for the one-form symmetry by (B.5):

$$Z^{\text{iELTO}}[\rho_\star + z] = Z^{\text{iELTO}}[\rho_\star] e^{-\frac{\pi i}{2} \int q_{\rho_\star}(z)}, \tag{58}$$

where $z \in H^2(M, \mathbb{Z}_2)$ classifies different background $\rho_2$ by its two-holonomy $\oint z$. It is a background two-form gauge field with the gauge transformation $z \to z + d\lambda$ that leaves the two-holonomy $\oint z$ invariant.

In the presence of a boundary, the partition function is not well-defined. In particular, the second term[23] in the effective action

$$-\frac{\pi i}{2} \int q_{\rho_\star}(z) \tag{59}$$

is not gauge invariant under a one-form background gauge transformation $z \to z + d\lambda$. It implies that the boundary of iELTO also realizes the one-form symmetry in an anomalous way. In particular, the charge of the one-form symmetry is not the worldline of a boson, but that of an anti-semion.

The single anti-semion theory has a $\mathbb{Z}_2$ one form symmetry that is anomalous. To see why the anomaly can be canceled by the bulk, let us take the spacetime manifold to be orientable for simplicity, then there is a canonical $\rho_\star = 0$, and the effective action (59) can be written as

$$-\int \frac{1}{2\pi} YY, \tag{60}$$

where we describes the $\mathbb{Z}_2$ two-form $z$ as a $U(1)$ background two-form field $Y \sim \pi z$ with $\mathbb{Z}_2$ holonomy $\oint Y = 0, \pi \bmod 2\pi$. The anti-semion theory can be expressed as a $U(1)_{-2}$ Chern-Simons theory with $U(1)$ gauge field $u$. It has $\mathbb{Z}_2$ one-form symmetry generated by $e^{i\oint u}$ which is the worldline of the anti-semion. The theory coupled to background gauge field $Y$ for the one-form symmetry as

$$-\int_{\text{boundary}} \frac{2}{4\pi} u du + \frac{2}{2\pi} u Y, \tag{61}$$

where $\oint Y = 0, \pi$ has $\mathbb{Z}_2$ valued holonomy. The one-form symmetry transforms $u \to u - \lambda$, $Y \to Y + d\lambda$, which transforms the Wilson line as $e^{i\oint u} \to e^{i\oint u} e^{-i\oint \lambda}$. The action is not invariant but changes by

$$\int_{\text{boundary}} \frac{2}{4\pi} \lambda d\lambda + \frac{2}{2\pi} \lambda Y. \tag{62}$$

This change on the boundary can be compensated by the transformation of the bulk term (60) [39], which is gauge invariant on a closed manifold but on an open manifold it changes by the opposite of the boundary term (62) such that the bulk-boundary system is gauge invariant.

---

[23]On an orientable manifold, the second term is the same as the effective action of the $\mathbb{Z}_2$ one-form symmetry SPT phase with $p = 3$ in [37] (or $m = 3$ in [38]).

This means that the anomaly on the boundary (61) is cancelled by the bulk (60) so the bulk system and boundary semion together couple to the background $Y$ consistently *i.e.* together have anomaly-free one-form symmetry.

The SPT phase distinguishes the half-braiding of the generator of the one-form symmetry on the boundary. Although semion and anti-semion have the same mutual braiding, only anti-semion can realize the required anomaly as discussed above. Indeed, if the sign of (60) is the opposite, then it can only cancel the anomaly of boundary semion theory instead the anti-semion theory, but not the other way around.

Another way to see the boundary has an anti-semion is by considering loops ending on the boundary. The intersection number of the open surfaces in 3+1d equals the linking number of the bounding loops on the 2+1d boundary. To see this, consider two closed loops $\gamma, \gamma'$ on the boundary, which we take to be $\mathbb{R}^3$, with $\gamma'$ being the boundary of some surface $\Sigma'$. The linking number of $\gamma, \gamma'$ on the boundary is

$$\text{link}(\gamma, \gamma') = \int_{3d} \delta_{3d}(\gamma)^{\perp} \delta_{3d}(\Sigma')^{\perp}, \tag{63}$$

where $\delta_{3d}(\gamma)^{\perp}$ is the Poincaré dual of $\gamma$ on the boundary and it is a delta function two-form that restricts the integral to $\gamma$, and similarly for $\delta_{3d}(\Sigma')^{\perp}$. Thus the integral computes the intersection number of $\gamma, \Sigma'$. Next, we can write it as a bulk integral by introducing bulk coordinate $z$,

$$\text{link}(\gamma, \gamma') = \int_{4d} \delta_{3d}(\gamma)^{\perp} \delta_{3d}(\Sigma')^{\perp} \delta(z)dz = \int_{4d} \delta_{4d}(\Sigma)^{\perp} \delta_{4d}(\Sigma')^{\perp} = \int_{4d} \delta_{4d}(\Sigma)^{\perp} \delta_{4d}(\Sigma'')^{\perp}, \tag{64}$$

where $\Sigma$ is an open surface in the bulk that intersects the $z = 0$ boundary by $\gamma$, and $\delta_{4d}(\Sigma')^{\perp} = \delta_{3d}(\Sigma')^{\perp} \delta(z)dz$. Then we can replace $\Sigma'$ by another surface $\Sigma''$ that extends in the bulk and ends on the boundary at locus $\gamma'$ without changing the integral, since $\Omega = \Sigma' \cup \overline{\Sigma''}$ is a closed surface, and $\int_{4d} \delta_{4d}(\Sigma)^{\perp} \delta_{4d}(\Omega)^{\perp} = 0$ for bulk with trivial topology. The last expression in (64) equals the intersection number of the surfaces $\Sigma, \Sigma''$ that bounds the curves $\gamma, \gamma'$ on the boundary. Thus when the world histories of loop intersect in the 3+1d bulk, the loops braid on the 2+1d boundary. This can also be understood from the following world history of two loops that intersect once: first, two loops are created from two points in 3+1d, they then move toward each other until crossing each other and become linked, and remain linked as they move to the boundary. Since the intersection of loop world history produces a minus sign, this implies that the braiding of loop in 2+1d also produces a minus sign, and the loop on the boundary represents a semion or anti-semion. (See Figure 1 for an illustration).

### 3.5.2 Boundary chiral central charge and thermal Hall property

Let us consider the partition function on orientable manifold,[24] and we turn off the background for the $\mathbb{Z}_2$ one-form symmetry. In this fixed background field, the partition function reduces to [13, 35, 36]

$$\exp\left(i\pi\frac{\sigma}{4}\right) = \exp\left(\frac{i}{96\pi}\int \text{Tr}\, R \wedge R\right), \tag{65}$$

where $\sigma$ is the signature of the manifold. On a manifold with boundary, the term $\int \frac{1}{96\pi}\text{Tr}R \wedge R$ can be rewritten as a gravitational Chern-Simons term $2\text{CS}_{\text{grav}} = \frac{1}{96\pi}\text{Tr}\left(\omega d\omega + \frac{2}{3}\omega^3\right)$ with

---

[24]Although we study the boundary property using the partition function on orientation manifold *i.e.* with a special choice of background gravity geometry, the same physics can in principle be obtained from the partition function on arbitrary manifolds.

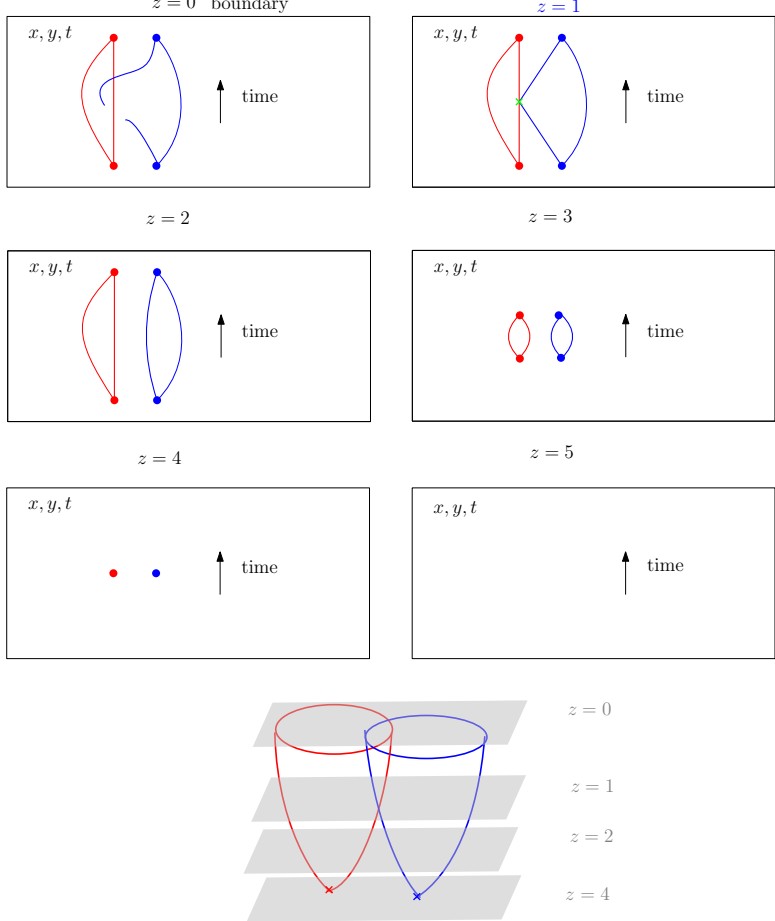

Figure 1: Slices of the two open surfaces that intersect in 3+1d bulk and end on the boundary ($z = 0$) by two linked loops. The slices are taken at different bulk coordinate $z \geq 0$. The loops on the boundary ($z = 0$) represent creating and annihilating two pairs of particles and braiding two of them in the time evolution. The intersection point in 3+1d bulk is denoted by the green dot at the slice $z = 1$.

spin connection $\omega$ on the boundary, which implies chiral central charge $c = -1$ on the boundary. This boundary chiral central charge is indicative of a boundary thermal Hall conductance. In general, a 3+1d bulk gravitational response gravitational response described by $-\frac{c}{96\pi} \int \text{Tr} R \wedge R$ gives rise to a 2+1d boundary gravitational Chern-Simons response with chiral central charge $c$. In a purely 2+1d system, the relation between the gravitational Chern-Simons response and the chiral central charge of the 1+1d conformal field theory on the 1+1d edge of system leads to the relation between the chiral central charge $c$ and the thermal Hall conductance

$$\kappa_{xy} = c, \tag{66}$$

where $\kappa_{xy}$ here is given in unit of $\pi^2 k_B^2 T/(3h) = \pi k_B^2 T/(6\hbar)$ with $T$ the temperature [40].[25] Here, even though we will mainly focus on a 3+1d system with a gravitational Chern-Simons response on its boundary, we assume that the relation between $\kappa_{xy}$ and $c$ still holds. Moreover, the boundary gravitational Chern-Simons response is also related to the contact term in the stress-tensor two-point function in 2+1d boundary [41].

In an ordinary bosonic SPT state in 3+1d, the time-reversal symmetry forces the chiral

---

[25]For instance, a boundary theory in 2+1d is the anti-semion theory, which has chiral central charge (or framing anomaly) $c = -1$, and the anti-semion theory can have 1+1d chiral edge boson theory with $c_L - c_R = -1$.

central charge on the boundary to be a multiple of 4 [18], where the time-reversal transformation produces a boundary $E_8$ state that can be compensated by the transformation of the bosonic bulk invertible phase with time-reversal symmetry and with effective action $\pi \int w_2^2$. On the other hand, here we find the boundary of the system without fermionic particles can have boundary chiral central charge that is an integer not a multiple of 4. This provides an invariant that distinguishes the exotic loop phase from the usual bosonic SPT phases.

The iELTO phase is time-reversal invariant, yet it has non-vanishing thermal Hall conductance due the the chiral central charge on the 2$d$ boundary. To probe the chiral central charge that generates the thermal Hall conductance, which is a gravitational response, let us place the system on a curved manifold that is orientable. In general, such manifolds can be non-spin. The time-reversal symmetry is the complex conjugation combining the map (21). The map (21) transforms the chiral central charge on the 2+1d boundary in the following way. The boundary of the iELTO has anomalous $\mathbb{Z}_2$ one-form symmetry, corresponds to the bulk SPT phase which on an orientable manifold is given by

$$-\frac{\pi}{2}\int \mathcal{P}(\rho_2). \tag{67}$$

Additional background $w_2$ for the one-form symmetry from the shift (21) produces an additional bulk action[26]

$$-\frac{\pi}{2}\int \mathcal{P}(w_2) = \frac{1}{48\pi}\int \mathrm{Tr}\,R\wedge R \quad \mathrm{mod}\ 2\pi\mathbb{Z}. \tag{68}$$

The right hand side of (68) can be written as a gravitational Chern-Simons term on the boundary, and it contributes additional thermal Hall conductivity $\Delta\kappa_{xy} = -2$ in the unit of $(\pi k_B)^2 T/3h$.[27] Then we find the modified time-reversal symmetry transforms the thermal Hall conductivity $\kappa_{xy}$ by

$$\mathcal{T}: \quad \kappa_{xy} \to -\kappa_{xy} - 2. \tag{69}$$

Thus the time-reversal invariant value is $c = \kappa_{xy} = -1$.

Consistently, since the anti-semion on the boundary has chiral central charge $c = -1$, it is symmetric under the modified time reversal symmetry. The ordinary time-reversal maps it to a semion, and after the modified time-reversal (21) the spin shifts by $1/2$ and it returns to the anti-semion, invariant under the time-reversal transformation.

### 3.5.3 A gapless boundary state with extended symmetry

We will discuss a possible 2+1d boundary state of the iELTO with local fermion particles, which are transformed under the $\mathbb{Z}_2$ fermion parity 0-form symmetry $\mathbb{Z}_2^{(0)}$, generated by $(-1)^F$. As discussed in Section 3.1.5, the two-group spacetime symmetry is extended on the boundary by the $\mathbb{Z}_2^{(0)}$ fermion parity symmetry to become the direct product of the $\mathbb{Z}_2$ one-form symmetry and the double-covering of Lorentz symmetry that satisfies $\mathcal{T}'^2 = (-1)^F$ (we used the notation $\mathcal{T}'$ to distinguish with the modified time-reversal symmetry $\mathcal{T}$ of the iELTO with the action (21)):

$$1 \to \mathbb{Z}_2^{(0)} \to \mathbb{G}_{\mathrm{boundary}}^{(2)} = \mathbb{Z}_2^{(1)} \times Pin^+(3) \to \mathbb{G}^{(2)} = \mathbb{Z}_2^{(1)} \times_{w_1 w_2} O(3) \to 1, \tag{70}$$

---

[26]We used the identity $-\mathcal{P}(w_2) = p_1 + 2(w_1^4 + w_1 w_3)$ mod 4, which follows from $\mathcal{P}(w_2) = p_1 + 2(w_1 Sq^1 w_2 + w_4)$ mod 4 [42,43] and $Sq^1 w_2 = w_3 + w_1 w_2$ mod 2, and the fact that $w_1^4 + w_2^2 + w_4$, $w_1^2 w_2$ are trivial mod 2 [32]. The last two terms that depend on $w_1$ are non-zero only on unorientable manifolds with nonzero $w_1$, so they do not contribute to the thermal Hall effect and are omitted here. Here, $p_1$ is the first Pontryagin class $p_1 = \frac{1}{24\pi^2}\mathrm{Tr}\,R\wedge R$

[27]This is the thermal Hall conductance before adding time-reversal symmetry breaking perturbations such as in [44].

where $\mathbb{G}^{(2)}_{\text{boundary}}$ is the symmetry of the boundary and we note that the 2+1d boundary only has $O(3)$ subgroup of the $O(4)$ Lorentz symmetry in the 3+1d bulk. We also emphasize that the $\mathbb{Z}_2$ fermion parity 0-form symmetry only acts on the boundary. Hence, the local fermions only reside on the boundary of the iELTO phase but not the bulk.

A possible gapless boundary state is $U(1)$ gauge theory with a single Dirac fermion of charge two, together with 7 decoupled neutral massless Dirac fermions. The theory has $\mathbb{Z}_2$ one-form symmetry that transforms the Wilson line of $U(1)$ charge one, while the Wilson line of even charges are screened by the charge-two fermion. The theory also has ordinary time-reversal symmetry $\mathcal{T}'$ that satisfies $\mathcal{T}'^2 = (-1)^F$. The theory has $\mathbb{Z}_2$ fermion parity symmetry $(-1)^F$ that transforms a spin-1/2 local operator given by the monopole operator, which is dressed with the charge-two fermion field to be gauge-invariant, and the fermion zero modes make the monopole operator acquire half-integer spin [45, 46]. The fermion parity symmetry also transforms the 7 decoupled massless Dirac fermions.

There are several ways to see the theory can be a boundary state of the iELTO phase. The $\mathbb{Z}_2$ one-form symmetry has the required anomaly for the theory to be a boundary of the iELTO phase, where the anomaly can be described by the part of the bulk effective action (59) that depends on the background of the one-form symmetry [46].

Another way to see the theory can be a boundary is using the particle/vortex duality. The $U(1)$ gauge theory with a single Dirac fermion of charge two has a dual description given by a free Dirac fermion and a decoupled anti-semion-fermion theory: [46]

$$U(1)_0 + \text{charge-two fermion} + 7 \text{ Free Dirac fermions} \longleftrightarrow 8 \text{ Free Dirac fermions} + U(1)_{-2}, \tag{71}$$

where $U(1)_0$ means that the $U(1)$ gauge theory has vanishing Chern-Simons level. The free Dirac fermions on the right hand side are purely on the boundary. This makes manifest that the $U(1)$ gauge theory with a single charge-two Dirac fermion can be a boundary theory of the iELTO phase. Following (71), the dual theory is given by the anti-semion theory stacked with 8 free massless Dirac fermion. More precisely, the presence of the free Dirac fermions imply that the anti-semion theory should be viewed as the anti-semion-fermion TQFT. From the dual theory perspective, the boundary theory has the following symmetries

- $\mathbb{Z}_2$ one-form symmetry that acts only on the anti-semion.

- Fermion parity symmetry $(-1)^F$ that transforms the free fermions.

- $O(16)$ flavor symmetry that rotates the 16 Majorana components of the Dirac fermions. We will not discuss this symmetry.

- Time reversal symmetry $\mathcal{T}'^2 = (-1)^F$ which acts on the Dirac fermion as $\mathcal{T}'\Psi(t) = \gamma^0 \Psi(-t)^*$. It also acts on the anti-semion-fermion theory. It is part of the $Pin^+(3)$ symmetry.

- Modified time reversal symmetry $\mathcal{T}^{\text{total}}$,

$$\mathcal{T}^{\text{total}}\Psi(t) = \mathcal{T}'\Psi(t) = \gamma^0 \Psi(-t)^*, \tag{72}$$

$$\mathcal{T}^{\text{total}}(\text{anti-semion}) = \mathcal{T}(\text{anti-semion}). \tag{73}$$

There is no mixed anomaly between the fermion parity symmetry and the modified time-reversal symmetry, and this ensures the fermion parity symmetry does not act in the bulk. In fact, the 16 free Majorana fermions can be removed by a deformation on the boundary that preserves both the modified time-reversal symmetry and fermion parity symmetry. Then after the boundary deformation, the boundary theory becomes the trivial fermionic SPT phase stacked with the anti-semion theory; it is thus manifestly a boundary state of the iELTO phase.

These symmetries have counterparts on the other side of the duality *i.e.* $U(1)$ gauge theory with charge-two fermion, together with 7 free Dirac fermions.

- $\mathbb{Z}_2$ one-form symmetry that acts on the $U(1)$ Wilson lines of odd charges.

- $O(14)$ flavor symmetry that transforms the 14 Majorana components of the 7 free Dirac fermions.

- $O(2) = U(1) \rtimes \mathbb{Z}_2$ symmetry that combines the $U(1)$ magnetic symmetry and the $\mathbb{Z}_2$ charge conjugation symmetry. The $O(2) \times O(14)$ symmetry is enhanced to $O(16)$ symmetry at low energy according to the duality.

- Time-reversal symmetry $\tilde{\mathcal{T}}'^2 = (-1)^F$

$$\tilde{\mathcal{T}}'\tilde{\Psi}_{q=2}(t) = \gamma^0 \tilde{\Psi}_{q=2}(-t), \quad \tilde{\mathcal{T}}'\tilde{\Psi}^I_{q=0}(t) = \gamma^0 \tilde{\Psi}^I_{q=0}(-t)^*, \tag{74}$$

where the subscript $q = 2, q = 0$ denote the charge two fermion and the neutral free Dirac fermions, with $I = 1, 2, \cdots, 7$.

- The gauge theory description also has modified time-reversal symmetry.

**Bosonic boundary from gauging fermion parity symmetry**   Another boundary, which does not have the fermion parity symmetry, can be obtained by gauging the fermion parity symmetry in the previous boundary theory. The new boundary theory is given by $U(1) \times \mathbb{Z}_2$ gauge theory without Chern-Simons term, where one massless Dirac fermion has charge $(2, 1)$ under the $U(1) \times \mathbb{Z}_2$ gauge group and the other 7 massless Dirac fermions have charge $(0,1)$. The basic monopole operator for the $U(1)$ gauge field, which is a fermion, now attaches to the $\mathbb{Z}_2$ Wilson line for the operator to be gauge invariant, and thus it is no longer a local operator. The monopole operator of magnetic charge two, which is a boson, remains a gauge-invariant local operator. At low energies, the theory flows according to the duality (71) to the tensor product of the $U(1)_{-2}$ anti-semion TQFT and the $\mathbb{Z}_2$ gauge theory without Chern-Simons term [47] coupled to 16 massless Majorana fermions. The latter sector can be gapped out while preserving the time-reversal symmetry, resulting in a $\mathbb{Z}_2$ gauge theory with Kramers singlet bosonic magnetic particle and it is free of time-reversal symmetry anomaly. The one-form symmetry of the two-group only acts on the anti-semion TQFT, and thus the $\mathbb{Z}_2$ gauge theory sector does not contribute to the anomaly of the two-group symmetry and can be gapped out, for example, by condensing the magnetic particle of the $\mathbb{Z}_2$ gauge theory.

## 3.6   Microscopic model: $SO(3)_-$ gauge theory

In this section we will describe a gauge theory in 3+1d that has the spacetime two-group symmetry, and at low energy it flows to the invertible exotic loop topological order. It is the $SO(3)_-$ theory with $\theta = 0$ in the notation of [48], where the subscript $-$ indicates that it has a non-trivial discrete theta parameter (given by (75) with $p = 1$). The theory is also equivalent to $SO(3)_+$ theory with $\theta = 2\pi$, where the subscript $+$ indicates that it has zero discrete theta parameter.[28] This UV model on a spin manifold gives essentially the same physics as the combination of the $\mathbb{Z}_2$ one-form symmetry SPT phase that belongs to the class $m = 3$ in the $\mathbb{Z}_4$ classification [37, 38] (whose partition function is given by (E.1)) and the $\nu = 2$ class DIII topological superconductor discussed in Subsection 3.7. The difference is that the model

---

[28]On spin manifolds, the discrete theta angle has the identification $p \sim p + 2$, and thus the discrete theta angles reduce to only two possible values as discussed in [48]. Here, we consider the theory on general non-spin manifolds and distinguish four different discrete theta angles $p = 0, 1, 2, 3 \mod 4$.

described by the $SO(3)_-$ gauge theory is well-defined on both spin and non-spin manifolds. So in this subsection, we will focus on the model on a non-spin manifold.

The discrete theta parameter can be described as follows. The $SO(3)$ gauge theory can be obtained from $SU(2)$ gauge theory by gauging the center electric $\mathbb{Z}_2$ one-form symmetry. Denote the two-form gauge field by $b$, which can be identified with the second Stiefel-Whitney class of the $SO(3)$ gauge field. The $SO(3)$ gauge theory has additional discrete theta term:

$$p\frac{\pi}{2}\int q_{\rho_2}(b). \tag{75}$$

Thus the theory has the following action in the Euclidean signature:

$$\int -\frac{1}{2g_{YM}^2}\mathrm{Tr}\, F_{\mu\nu}F^{\mu\nu}+\frac{i\theta}{32\pi^2}\epsilon^{\mu\nu\lambda\rho}\mathrm{Tr}\, F_{\mu\nu}F_{\lambda\rho}+ip\frac{\pi}{2}q_{\rho_2}(b), \tag{76}$$

for $SO(3)$ field strength $F$, and the second Stiefel-Whitney class of the $SO(3)$ bundle is $b$. The $SO(3)$ gauge theory with $p=0$ is denoted as $SO(3)_+$ and the $SO(3)$ gauge theory theory with $p=1$ is denoted as $SO(3)_-$. For the $SO(3)_-$ theory on a oriented manifold, we can rewrite the discrete theta terms as $\frac{\pi}{2}\int \mathcal{P}(b)+\pi\int \rho_2\cup b$ which indicates that $\rho_2$ can be understood as the background of the magnetic $\mathbb{Z}_2$ one-form symmetry generated by $e^{i\oint b}$ in the $SO(3)_-$ gauge theory. As shown later, the $SO(3)_-$ theory in fact has the $\mathbb{Z}_2^{(1)}\times_{w_1w_2}O(4)$ two-group symmetry. Hence, the background field $\rho_2$ should be more generally viewed as the Wu structure of degree three, which enables us to define the theory on general unorientable spacetime.

$SO(3)$ or $SU(2)$ gauge theory has line operators $(q_e, q_m)$ labeled by the weight $q_e$ and coweight $q_m$ of $\mathfrak{su}(2)$, which represent the electric and magnetic (GNO) charge which are all integers [49]. The lattice of line operators depend on the gauge group and the discrete theta parameter $p$ as discussed in [48,50]. The magnetic $\mathbb{Z}_2$ one-form symmetry acts on the lines with odd $q_m$. The lines $(q_e, q_m), (q'_e, q'_m)$ have mutual statistical Berry phase $e^{i(q_e q'_m - q_m q'_e)\Omega/4}$, where $\Omega$ is the solid angle subtended by the trajectory of one particle at a fixed distance with the other particle placed at the origin, which is equivalent to the Berry phase for a particle in a constant-magnitude electric and magnetic fields pointing in varying radial directions. Note that $(q_e q'_m - q_m q'_e)/4$ is also the angular momentum stored in the gauge field [51] in unit of $\hbar$. The self-statistics of the spectrum $(q_e, q_m)$ for the $SO(3)$ gauge theory with $\theta=0$ and discrete theta parameter $p$ is

$$h(q_e, q_m)=\frac{q_m(q_e-pq_m)}{4}. \tag{77}$$

Based on Ref. [50], the genuine line operators of the $SO(3)_+$ and the $SO(3)_-$ theories (both with $\theta=0$) are marked as the green dots shown in Figure 2. The letters "$b$" and "$f$" indicates the bosonic and fermionic statistics of the corresponding line.

### 3.6.1 Low energy excitations in $SO(3)_\pm$ theories

Consider the basic electric line $E$ with charge $(1,0)$ and the basic 't Hooft line $M$ with charge $(0,1)$. The electric line transforms under the center one-form symmetry in $SU(2)$ gauge theory, and the symmetry is gauged in the $SO(3)$ gauge theory with two-form gauge field $b$. Thus the electric line is not gauge invariant, the gauge invariant combination is

$$E e^{\pi i \int_\Sigma b}, \tag{78}$$

where the line is supported on the boundary of the surface $\Sigma$. Thus the electric line $(q,0)$ with odd $q$ is not a genuine line operator in the $SO(3)$ gauge theory. The basic magnetic line $M$ with charge $(0,1)$ carries unit flux of $\oint b$. In the theory with discrete theta parameter $p$, $M$ is

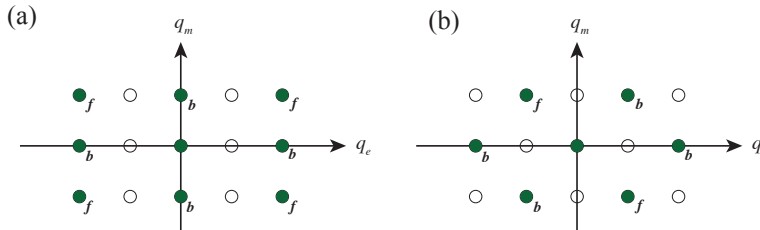

Figure 2: The genuine line operators of the $SO(3)_+$ and the $SO(3)_-$ theories (both with $\theta = 0$) are marked as the green dots in the integer lattice $(q_e, q_m)$. The letters "$b$" and "$f$" indicates the bosonic and fermionic statistics of the corresponding line.

also not invariant under the gauged one-form center symmetry if $p$ is odd, and it needs to be attached with a Wilson surface

$$M e^{p\pi i \int_\Sigma b}, \tag{79}$$

where the line is supported on the boundary of the surface $\Sigma$. Therefore, in the theory with even $p$, the electric line (78) is not a genuine line, since it depends on the surface by $e^{\pi i \int b}$, but the magnetic line (79) is a genuine line. The dyon, given by their fusion, is also not a genuine line. In the theory with odd $p$, both the electric line (78) and the magnetic line (79) are not genuine lines, but the dyon, given by their fusion, is a genuine line, since the surface dependence cancels $e^{\pi i \int b} e^{p\pi i \int b} = 1$ for odd $p$. This reproduces the spectrum in $SO(3)$ gauge theory with discrete theta angle $p$ [48, 50]. Again, the spectrum for discrete theta angle $p = 0, 1$ is shown in Figure 2. In the following we will discuss the low energy behavior of the microscopic spectrum in the $SO(3)$ gauge theory.

**Genuine line operators** For the $SO(3)_+$ theory with $p = 0$ (and $\theta = 0$), in the phase where the magnetic monopole $(0, 2)$ condenses, the monopole $(0, \pm 1)$ is deconfined. The low-energy physics describes the $\mathbb{Z}_2$ gauge theory. In contrast, in the $SO(3)_-$ theory with $p = 1$ (and $\theta = 0$), there is no deconfined particles after the condensation of the magnetic monopole $(0, 2)$. The low-energy physics describes the iELTO phase. The line operators of the confined dyons $(\pm 1, \pm 1)$ are charged under the one-form symmetry associated with the background $\rho_2$. Two of these dyons are bosons while the other two are fermions. They are pairwise related by time-reversal symmetry: $\{(-1, 1), (1, 1)\}$ and $\{(1, -1), (-1, -1)\}$, with each pair contains a boson and a fermion. [29] In fact, the theory with the $p = 1$ lattice has the spacetime two-group symmetry, while the theory with $p = 0$ lattice has the ordinary time-reversal symmetry.[30]

**Non-genuine line operators** In the theory $SO(3)_+$ with $p = 0$ and $\theta = 0$, the boundary of the open surface $\Sigma$ of the surface operator $e^{i\pi \int_\Sigma b}$ can be the $(n, 0)$ operator with any odd $n$, which is indeed not a genuine line on the $p = 0$ lattice. In the theory $SO(3)_-$ with $p = 1$ and $\theta = 0$, the intersection of surfaces $\int b$ produces a minus sign. Thus the boundary of the open surface carries magnetic charge, and the only such non-genuine line operators are $(0, n)$ for any odd $n$. Note the dyons $(n, m)$ for even $n + m$, i.e. for $n, m$ both even or both odd, are genuine lines in the $SO(3)_-$ theory [48]. If we view the $SO(3)_-$ theory as the $SO(3)_+$ theory at $\theta = 2\pi$, the dyons in $SO(3)_-$ can be obtained, by increasing $\theta = 0$ to $\theta = 2\pi$, from the lines in

---

[29]We will focus on the vacuum where the monopole $(0, 2)$ condenses. We can also study the phase when the dyon $(2, 2)$ condense. The monopole $(0, 1)$ on the $p = 0$ lattice confines, while the dyon $(\pm 1, \pm 1)$ on $p = 1$ lattice is deconfined.

[30]This is consistent with the low energy $\mathbb{Z}_2$ gauge theory for $p = 0$ has ordinary time-reversal symmetry and not the two-group spacetime symmetry.

$SO(3)_+$ at $\theta = 0$ that have even electric charges. On the other hand, the pure magnetic lines $(0, n)$ in $SO(3)_-$ with odd $n$ are obtained from the dyon lines $(-1, n)$, which are not genuine lines in the $SO(3)_+$ theory with $\theta = 0$. These lines are attached to the surface $\int b$ as they carry odd electric charges. The lines that attached to surfaces in $PSU(N)$ gauge theory with discrete theta parameter are also discussed in [37].

**Phase transition at $\theta = \pi$**   We remark that as we tune $\theta$ in $SO(3)_+$ gauge theory between $\theta = 0$ and $\theta = 2\pi$, the theory interpolates between the $p = 0$ theory with low energy $\mathbb{Z}_2$ gauge theory, and the $p = 1$ theory, namely the $SO(3)_-$ theory, with low energy iELTO phase. There must be a phase transition in between the two phases, which is proposed to occur at $\theta = \pi$ [52].

### 3.6.2   Time-reversal symmetry

Under the ordinary time-reversal transformation, the line operator $(q_e, q_m)$ in $SO(3)$ gauge theory with $\theta = 2\pi$ (or $SO(3)_-$ in this paper) is mapped to $(-q_e, q_m)$ in $SO(3)$ gauge theory with $\theta = -2\pi$, since the electric field $F_{0i}^a$ flips sign while the magnetic field $F_{ij}^a$ stays invariant. In particular, the ordinary time-reversal symmetry transforms the bosonic dyon $(1, 1)$ to the bosonic dyon $(-1, 1)$, while the dyon $(-1, 1)$ in the $SO(3)_-$ is a fermion. Thus the theory is not invariant under the ordinary time-reversal symmetry.

Another way to see the ordinary time-reversal transform is not a symmetry is as follows. The ordinary time-reversal transformation flips the sign of the theta term, changing $\theta = 2\pi$ into $\theta = -2\pi$. They differ by a theta term with $\theta = 4\pi$. Such theta term $\theta = 4\pi$ is non-trivial, and thus the theory is not invariant under the ordinary time-reversal symmetry. To see the theta term $\theta = 4\pi$ is a non-trivial topological term, we can compare the operator spectrum in the theories with $\theta = 4\pi$ and $\theta = 0$. The statistics of the particle with odd magnetic charge differ by spin 1/2. To understand this effect, we note that in the presence of magnetic flux $m$, an electric charge $q$ (in the unit of the minimal electric charge in the theory) has canonical angular momentum shifted by $qm/2$ [53]. The pure monopole $(0, 1)$ at $\theta = 4\pi$ comes from a dyon at $\theta = 0$ due to the Witten effect [51], and the dyon $(-2, 1)$ at $\theta = 0$ is a fermion due to the shifted angular momentum, and thus the monopole at $\theta = 4\pi$ is also a fermion [54], while the monopole $(0, 1)$ at $\theta = 0$ is a boson.[31]

However, as we will see, the theory has a modified time-reversal symmetry where $(q_e, q_m)$ and $(-q_e, q_m)$ transform into one another. In particular, the bosonic dyon $(1, 1)$ and the fermionic dyon $(-1, 1)$ form a doublet under the modified time reversal symmetry.

**Modified time-reversal symmetry**   The ordinary time-reversal changes $\theta = 2\pi$ to $\theta = -2\pi$ (or equivalently, $p = 1$ to $p = -1$). The difference is

$$\frac{\pi}{2} \int q_{\rho_2}(b) - \left( -\frac{\pi}{2} \int q_{\rho_2}(b) \right) = \pi \int b \cup b = \pi \int b \cup w_2 = \frac{\pi}{2} \int q_{\rho_2 + w_2}(b) - \frac{\pi}{2} \int q_{\rho_2}(b). \tag{80}$$

The operator $\oint b$ generates the magnetic one-form symmetry that transforms the line operators that create the magnetic charges. The background for the one-form symmetry is $\rho_2$. By comparing (80) and (75), we find that the time-reversal symmetry in addition to performing the ordinary time-reversal transformation that brings $\theta = 2\pi$ to $\theta = -2\pi$, must also act on

---

[31]Another way to understand this is noting that the theta term $\theta = 4\pi$ on a manifold with boundary gives rise to $SO(3)_1 = SU(2)_2/\mathbb{Z}_2$ Chern-Simons term, which has a magnetic monopole that carries an electric charge due to the Chern-Simons interaction and becomes a fermion by the same statistics shift $em/2$. This monopole comes from the bulk monopole, so the bulk monopole is also a fermion.

the background field $\rho_2$ to compensate the difference $\theta = 4\pi$:

$$\text{Modified time-reversal } \mathcal{T}: \quad \rho_2 \rightarrow \rho_2 + w_2. \tag{81}$$

This additional transform also has the effect of changing the spin of the particles for the lines carrying odd magnetic charge by $1/2$. The modified time-reversal symmetry indicates that the UV $SO(3)_-$ gauge theory (or $SO(3)$ gauge theory with $\theta = 2\pi$) has the same spacetime two-group symmetry as that of the iELTO phase.[32] In fact, we will argue that the low energy physics of the $SO(3)$ gauge theory with the discrete theta parameter $p = 1$ is the iELTO phase.

### 3.6.3 iELTO phase

At low energy when the $SU(2)$ or $SO(3)$ gauge field dynamics confines, the $SO(3)_-$ theory is described purely by the two-form $\mathbb{Z}_2$ gauge theory with the action (34), which also describes the iELTO phase as discussed in Section 3. Thus the $SO(3)_-$ provides a microscopic description of the iELTO phase that has time-reversal symmetry. The basic 't Hooft line $(0,1)$, which is a non-genuine line in the UV, when terminates on the $2+1$d boundary, carries an antisemion, with self-statistics $h(0,1) = -\frac{1}{4}$ according to (77). Thus the $(0,1)$ line is the loop in the iELTO phase at low energy.[33]

### 3.6.4 A phase transition with spacetime two-group symmetry

In this section we discuss a theory describing a quantum phase transition in 3+1d with the spacetime two-group symmetry, and it has a trivial symmetric phase and the iELTO phase, similar to the theory of massless Majorana fermion in 1+1d that has the trivial symmetric phase and the Kitaev's chain invertible fermionic phase.

Consider $SO(3)$ gauge theory with $\theta = 2\pi$ and two scalars in the vector representation, and the scalars have a quartic potential,

$$\frac{1}{4\pi}\text{Tr } (F \wedge F) + \frac{1}{2g^2}\text{Tr } (F \wedge \star F) + \sum_{i=1,2}(D_a \phi^i)^2$$
$$- M_i^2(\phi^i)^2 - \lambda_i(\phi^i)^4 - \lambda_{12}(\phi^1)^2(\phi^2)^2 - \lambda'(\phi^1 \cdot \phi^2)^2, \tag{82}$$

where $a$ is the $SO(3)$ gauge field with field strength $F$, and $\phi^i$ with $i = 1, 2$ are two real scalars in the vector representation. We will take the couplings $\lambda_i, \lambda_{12}, \lambda' > 0$, with $\lambda_{12}$ small compared with $\lambda_i, \lambda'$.

The theory has $\mathbb{Z}_2$ magnetic one-form symmetry (denoted as $\mathbb{Z}_2^{(1)}$ in Figure 3) and time-reversal symmetry $\mathcal{T}$ that combine into the spacetime two-group symmetry, as in the pure $SO(3)$ gauge theory with $\theta = 2\pi$ discussed previously. The scalars $\phi^{i=1,2}$ are neutral under these symmetries. We also impose a $\mathbb{Z}_2$ flavor symmetry that exchanges the two scalars, which constrains $M_1 = M_2 = M$ and $\lambda_1 = \lambda_2 = \lambda$.

The theory has a relevant operator given by the mass term $M^2 \sum_i (\phi^i)^2$. Under the deformation by such operator, the theory has the following phases:

- $M^2 < 0$: the scalars $\phi^{1,2}$ both condense with their vacuum expectation values orthogonal to each other due to $\lambda'$. Hence, the gauge group $SO(3)$ is completely Higgsed, at low energy leading to a trivial symmetric phase.

---

[32]On the other hand, both the $SU(2)$ gauge theory with $\theta = 2\pi$ and the $SO(3)$ gauge theory with $\theta = 0$ (*i.e.* $SO(3)_+$) are invariant under the ordinary time-reversal, (their weight lattices are symmetric under $(q_e, q_m) \rightarrow (-q_e, q_m)$,) and thus they break the modified time reversal symmetry.

[33]We note that at low energy the particle excitations are confined due to monopole condensation, and they do not appear in the low energy description.

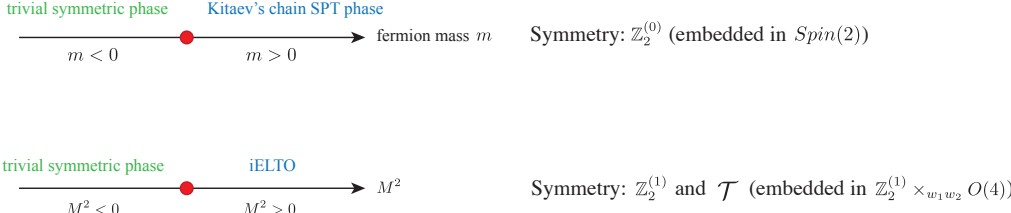

Figure 3: The upper diagram describes the phase transition of massless Majorana fermion in 1+1d with the fermion parity symmetry $\mathbb{Z}_2^{(0)}$ embedded in the full spacetime $Spin(2)$ symmetry. This phase transition separates the trivial phase and the Kitaev's chain invertible fermionic phase. The free Majorana fermion in 1+1d is also dual to the $\mathbb{Z}_2$ gauge theory with the discrete theta angle (6), and the gauge field couples to a real scalar that has a quartic potential (for a review, see *e.g.* (2.9) of [55], where the $\mathbb{Z}_2$ gauge field is dentoed by $s$, and the discrete theta angle is denoted by $\frac{\pi}{2} \int q_\rho(s) = \pi \int \mathrm{Arf}(s \cdot \rho) + \mathrm{Arf}(\rho)$ with spin structure $\rho$), which is the 1+1d analogue of the action (82). The mass square of the real scalar is identified with the fermion mass $m$ with appropriate sign. The lower diagram describes the phase transition with exotic loops, which has $\mathbb{Z}_2^{(1)}$ one-form symmetry and the (modified) time-reversal symmetry $\mathcal{T}$ that are both embedded in the spacetime two-group symmetry $\mathbb{Z}_2^{(1)} \times_{w_1 w_2} O(4)$). The two sides of the phase transition are given by the trivial phase and the iELTO phase respectively. A proposed theory of this transition is given in (82). In both phase diagrams, the symmetry is unbroken on both sides of the phase transition, and thus the symmetry is also unbroken at the phase transition.

- $M^2 > 0$: the scalars do not condense, and at low energy the theory becomes pure $SO(3)$ gauge theory with $\theta = 2\pi$, which describes the iELTO phase.

Both sides of the phase transition are in the confined phase with a symmetric vacuum and unbroken one-form symmetry, so the symmetry is also unbroken at the transition. Moreover, since the two sides are different SPT phases for $\mathbb{Z}_2$ one-form symmetry, the phase transition must contain deconfined excitations to account for the difference. It is thus a plausible deconfined quantum critical point with the spacetime two-group symmetry. The phase diagram of the theory is sketched in the lower part of Figure 3.

If we do not impose the $\mathbb{Z}_2$ symmetry that exchanges the two scalars, we can turn on a negative mass square for one of the two scalars, while the other scalar remains massless. Then the theory flows to the $U(1)$ gauge theory with $\theta = \pi$ coupled to a scalar of charge one. The theory also has two-group spacetime symmetry, where the magnetic one-form symmetry is enhanced from $\mathbb{Z}_2$ to $U(1)$.[34] If furthermore the remaining scalar has positive mass square, then the theory flows to the free $U(1)$ gauge theory with $\theta = \pi$, which has emergent $U(1)$ electric and magnetic one-form symmetries, both are spontaneously broken. Alternatively, we can also turn on a positive mass square for one of the two scalars, while the other scalar remains massless, then the theory flows to $SO(3)$ gauge theory with $\theta = 2\pi$ coupled to a scalar in the vector representation. The theory also has the spacetime two-group symmetry.

---

[34]The background $B$ for the $U(1)$ magnetic one-form symmetry satisfies

$$dB = \pi w_1 w_2 \mod 2\pi\mathbb{Z}. \tag{83}$$

## 3.7 Comparison with other topological phases

Table 1: Comparisons of different phases and their boundary properties. The thermal Hall conductance $\kappa_{xy}$ is given in the unit of $\pi^2 k_B^2 T/(3h)$ where $T$ is the temperature. We also include examples of possible surface theories for the bulk SPT phases. The time-reversal $\mathcal{T}$ and one-form symmetry $\mathbb{Z}_2^{(1)}$ in the iELTO row are embedded in the two-group symmetry $\mathbb{Z}_2^{(1)} \times_{w_1 w_2} O(4)^{(0)}$, while the time-reversal symmetry $\mathbb{Z}_2^T$ and fermion parity 0-form symmetry $\mathbb{Z}_2^F$ in the middle three rows satisfy $\mathcal{T}'^2 = (-1)^F$ with $\mathcal{T}'$ the generator of the time-reversal symmetry. We remark that the expression of the effective action for the iELTO phase is valid only on orientable manifold, and in particular on closed manifold the gravitational part becomes trivial, just as the gravitational part of the effective action for topological superconductor with even $\nu$ on spin manifolds. Thus we cannot make comparison between the iELTO and the topological superconductor using the gravitational part in the above effective actions. In the last row, $B$ is the background for the $\mathbb{Z}_2$ one-form symmetry. $\mathcal{P}$ is the Pontraygin square operation. The boundary theory of the $m = 3$ $\mathbb{Z}_2^{(1)}$ SPT phase should break the ordinary time reversal symmetry and should at least include an anti-semion to account for the anomaly of the one-form symmetry. A double-semion boundary theory (with broken time-reversal symmetry) is constructed in [38] for the $m = 3$ $\mathbb{Z}_2^{(1)}$ SPT phase. In the last column from the left, the particles in $SO(3)_3 = SU(2)_6/\mathbb{Z}_2$ topological order can be labelled by the $SU(2)$ isospin $j = 0, 1, 2, 3$ with topological spins $0, \frac{1}{4}, \frac{3}{4}, \frac{1}{2}$ mod 1, denoted by $1, s, \tilde{s}, f$. The notation $s$ stands for semions, while $\tilde{s}, \bar{s}'$ stands for anti-semions, and $f$ denotes the local fermion. Note that here we refer to some of these particles as semions and anti-semions purely based on their topological spins $\frac{1}{4}$ and $\frac{3}{4}$. However, the semion $s$ and anti-semion $\tilde{s}$ in the $SO(3)_3$ TQFT are non-Abelian anyons while the semions $s$ in the semion TO and the anti-semion $\bar{s}'$ in the anti-semion TO are Abelian anyons.

| Bulk Phase | Symmetry | Effective action | Class | Boundary theories |
|---|---|---|---|---|
| iELTO | $\mathbb{Z}_2^{(1)} \times_{w_1 w_2} O(4)^{(0)}$ | $\frac{1}{96\pi} \int \text{Tr} R \wedge R$ $-\frac{\pi}{2} \int \mathcal{P}(\rho_2)$ | $\mathbb{Z}_8$ | anti-semion (time-reversal invariant) |
| $\nu = 1$ TSC | $\mathbb{Z}_2^F, \mathbb{Z}_2^T$ | $\frac{1/4}{96\pi} \int \text{Tr} R \wedge R$ | $\mathbb{Z}_{16}$ | Bdy I: gapless, $\kappa_{xy} = \frac{1}{4}$ [44, 56] Bdy II: $SO(3)_3 \times U(1)_{-2}$ TO $= \{1, s, \tilde{s}, f\} \times \{1, \bar{s}'\}$ |
| $\nu = 2$ TSC | $\mathbb{Z}_2^F, \mathbb{Z}_2^T$ | $\frac{1/2}{96\pi} \int \text{Tr} R \wedge R$ | $\mathbb{Z}_8 \subset \mathbb{Z}_{16}$ | Bdy I: gapless, $\kappa_{xy} = \frac{1}{2}$ Bdy II: semion-fermion TO $= \{1, s\} \otimes \{1, f\}$ |
| $\nu = 3$ TSC | $\mathbb{Z}_2^F, \mathbb{Z}_2^T$ | $\frac{3/4}{96\pi} \int \text{Tr} R \wedge R$ | $\mathbb{Z}_{16}$ | Bdy I: gapless, $\kappa_{xy} = \frac{3}{4}$ Bdy II: $SO(3)_3$ TO $= \{1, s, \tilde{s}, f\}$ |
| $m = 3$ $\mathbb{Z}_2^{(1)}$ SPT | $\mathbb{Z}_2^{(1)}$ | $-\frac{\pi}{2} \int \mathcal{P}(B)$ | $\mathbb{Z}_4$ | at least an anti-semion (not time-reversal invariant) |

As discussed in Section 3.1.5, with additional fermion parity symmetry, the spacetime two-group symmetry in the iELTO phase is extended to be the direct product of $\mathbb{Z}_2$ one-form symmetry and $Pin^+(4)$ (it is the extension of $O(4)$ by the fermion parity symmetry such that the time-reversal symmetry squares to $(-1)^F$ i.e. class DIII). This implies that the iELTO phase, when stacked with additional transparent fermion (a decoupled sector representing a trivial class fermionic SPT phase with trivial effective action), becomes the product of a one-form symmetry SPT phase and a topological superconductor (TSc) in class DIII. Such phases have $\mathbb{Z}_{16} \times \mathbb{Z}_4$ classification [57]. We will investigate which phase that the iELTO becomes after

stacking with trivial fermionic SPT phase.

Let us compare the invertible exotic loop topological order and other $3 + 1$-dimensional SPT phases with relevant symmetries: topological superconductors with fermion parity 0-form symmetry and time-reversal symmetry (DIII class, such as the Helium 3B phase) and SPT phase with $\mathbb{Z}_2$ one-form symmetry. Their effective actions and examples of boundary states are listed in Table 1. We also include the classification these phases generate, for instance $\mathbb{Z}_8$ means eight copies of the phase results in a trivial phase.

Since a boundary state of the iELTO phase is the anti-semion theory, in the presence of transparent fermion particle on the boundary it becomes a boundary state for the $\nu = \pm 2$ TSc, where $\pm$ depends on the choice of $pin^+$ structure. Moreover, the line that creates an anti-semion generates a $\mathbb{Z}_2$ one-form symmetry with the anomaly described by $m = -1$ one-form symmetry SPT. Thus we find[35]

$$\text{iELTO} \times (\text{class } \nu \text{ TSc}) \longleftrightarrow (\text{class } (\nu + 2) \text{ TSc}) \times (m = -1 \text{ one-form symmetry SPT}). \quad (85)$$

More precisely, the one-form symmetry SPT phase on the right hand side is the $-1$ element in the $\mathbb{Z}_4$ part of the classification $\Omega^4_{pin^+}(B^2\mathbb{Z}_2) = \mathbb{Z}_{16} \times \mathbb{Z}_4$ [57].

More generally, by taking $\nu'$ copies of the iELTO phase we obtain $\mathbb{Z}_8$ classification of the SPT phase with the two-group symmetry.

$$(\text{iELTO})^{\nu'} \times (\text{class } \nu \text{ TSc}) \longleftrightarrow (\text{class } (\nu + 2\nu') \text{ TSc}) \times (m = -\nu' \text{ one-form symmetry SPT}). \quad (86)$$

This is also consistent with the discussion in Section 3.3 that four copies of the iELTO phase is a bosonic SPT phase with time-reversal symmetry, which has partition function $-1$ on $\mathbb{RP}^4$, the latter is also the difference between class $\nu$ and class $\nu + 8$ TSc.

## 4 Higher dimension generalization

In this section, we comment on the generalization of the discussions in previous sections to any even spacetime dimension $2n$. This gives an invertible phase with exotic $(n-1)$-membrane excitation that is charged under a $\mathbb{Z}_2$ $(n-1)$-form symmetry, and we call it invertible exotic higher topological order (iETO). The discussion is exactly parallel to the previous sections and we will not give a detailed discussion here.

The $n$-group symmetry can be understood as a non-trivial extension of the Lorentz symmetry $O(d)$ for $d = 2n$ spacetime dimension by the $(n-1)$-form symmetry

$$1 \to \mathbb{Z}_2 \to \mathbb{G}^{(n)} \to O(d) \to 1. \quad (87)$$

The $n$-group extension is specified by the Postnikov class, and for the $n$-group discussed here it is given by $\omega_{n+1} = v_{n+1} \in H^{n+1}(BO(d), \mathbb{Z}_2)$, where $v_{n+1} \in H^{n+1}(BO(d), \mathbb{Z}_2)$ is the $(n+1)$th Wu class. . We can denote the $n$-group by

$$\mathbb{G}^{(n)}[v_{n+1}] = \mathbb{Z}_2^{(n-1)} \times_{v_{n+1}} O(d)^{(0)}. \quad (88)$$

---

[35]The background $\rho_2$ can be expressed as $\rho_2 = w_1 \cup \eta + B$ for $pin^+$ structure $\eta$ that satisfies $w_2 = \delta\eta$, and $\delta B = 0$. Then changing the $pin^+$ structure by $\eta \to \eta + w_1$ changes $\rho_2$ by $\rho_2 \to \rho_2 + w_1^2$, and the action of the two-form gauge theory describing iELTO changes as, $\frac{\pi}{2}q_{\rho_2}(b)$, changes as

$$\frac{\pi}{2}q_{\rho_2}(b) \to \frac{\pi}{2}q_{\rho_2}(b) + \pi b \cup w_1^2 = \frac{\pi}{2}q_{\rho_2}(b) + \pi b \cup b = -\frac{\pi}{2}q_{\rho_2}(b) \mod 2\pi\mathbb{Z}, \quad (84)$$

where we used the Wu formula $b \cup (w_2 + w_1^2) = b \cup b$, and $w_2 = \delta\eta$ is trivial on $pin^+$ manifolds. Thus we can choose a convention of $pin^+$ structure such that the right hand side of (85) to be $\nu + 2$.

The associativity of the $O(d)$ spacetime symmetry transformations is modified: different ways of performing $n+1$ symmetry actions $h_1, \cdots, h_{n+1}$ of $O(d)$ differ by an action of $\mathbb{Z}_2^{(n-1)}$-form symmetry if $v_{n+1}(h_1, \cdots, h_{n+1})$ is non-trivial. When $n$ is even, the Wu class $v_{n+1}$ becomes trivial when the time-reversal symmetry is not present *i.e.* when the $O(d)$ symmetry is replaced by $SO(d)$ symmetry. In such a case, the $n$-group symmetry factorizes into the product of $\mathbb{Z}_2$ $(n-1)$-form symmetry and the $SO(d)$ symmetry. In contrast, when $n$ is odd, the $n$-group symmetry is non-trivial even in the absence of time-reversal symmetry.

The background of the $n$-group symmetry satisfies

$$d\rho_n = v_{n+1}, \tag{89}$$

where $\rho_n$ is the background for the $(n-1)$-form symmetry, and $v_{n+1}$ is the pullback of the $(n+1)$th Wu class $v_{n+1} \in H^{n+1}(BO(d), \mathbb{Z}_2)$ to the manifold. Equation (89) implies that the manifold is equipped with an $n$-group bundle, and $\rho_n$ plays the role of the $v_{n+1}$ Wu structure, *i.e.* Wu structure of degree $n+1$.

## 4.1 Invertible exotic higher topological order

An invertible phase with $n$-group symmetry is described by the following $n$-form $\mathbb{Z}_2$ gauge theory with the action

$$\frac{\pi}{2} \int q_{\rho_n}(b), \tag{90}$$

where $b$ is a dynamical $n$-form $\mathbb{Z}_2$ gauge field. $q$ is the quadratic function whose properties we summarize in Appendix B. For the action to be well-defined, $\rho_n$ is a $n$-form $\mathbb{Z}_2$ background that satisfies (89), and thus the theory has $n$-group symmetry. The theory is gapped and has a unique ground state on any space manifold by a similar argument as in (7), and it describes an invertible topological order with the $n$-group symmetry. The theory has $(n-1)$-dimensional membranes described by the holonomy of the $n$-form gauge field $b$, and it transforms under the $(n-1)$-form symmetry. When the worldvolume of the $(n-1)$-membrane intersects in spacetime, it contributes a sign to the correlation function.

The invertible phase described by the $n$-form gauge theory (90) has order 8, *i.e.* 8 copies of the theory gives the trivial phase, similar to the discussion in Section 3.3. Four copies of the theory differs from the trivial class by a bosonic SPT phase, the latter has effective action $\pi \int v_n^2$ with degree-$n$ Wu class $v_n$.

# 5 More exotic phases from "fermionization"

In fact, we could always obtain a phase with an exotic $(n-1)$-dimensional excitation in $2n$ spacetime dimensions, protected by the spacetime $n$-group, starting with a bosonic theory in $2n$ spacetime dimensions that has a non-anomalous $(n-1)$-form $\mathbb{Z}_2$ symmetry, where for even $n$ we require the theory also has time-reversal symmetry that does not mix with the $(n-1)$-form symmetry *i.e.* the time-reversal symmetry does not transform the background of the $(n-1)$-form symmetry. We will show this by deriving the partition function of the exotic phase. To begin with, consider such a bosonic theory $T$ with a $\mathbb{Z}_2$ $(n-1)$-form symmetry in $2n$ spacetime dimensions. In the presence of a background $\mathbb{Z}_2$ $n$-form gauge field $b$ associated with the $\mathbb{Z}_2$ $(n-1)$-form symmetry, the bosonic theory $T$ has a partition function denoted as $Z_T[b]$. We could gauge the $\mathbb{Z}_2$ $(n-1)$-form symmetry and obtain another bosonic theory $T/\mathbb{Z}_2$ whose partition function is given by

$$Z_{T/\mathbb{Z}_2}[x] = \frac{1}{|H^n(M, \mathbb{Z}_2)|^{1/2}} \sum_{b \in H^n(M, \mathbb{Z}_2)} Z_T[b] e^{\pi i \int b \cup x}, \tag{91}$$

where $b, x$ are the $n$-form $\mathbb{Z}_2$ gauge fields on $2n$-dimensional manifold $M$, with the gauge transformations $b \to b + d\lambda_b, x \to x + d\lambda_x$.[36] Note that the gauged bosonic theory $T/\mathbb{Z}_2$ has a dual $(n-1)$-form $\mathbb{Z}_2$ symmetry. We've made the dependence of its partition function $Z_{T/\mathbb{Z}_2}[x]$ on the background $n$-form $\mathbb{Z}_2$ gauge field $x$ explicitly.

Beginning with either bosonic theory, we can obtain a new class of theories with the spacetime $n$-group symmetry, by coupling it to the invertible exotic higher topological order described by (90). We first couple the theory to a background $\mathbb{Z}_2$ $n$-form gauge field $B$ using the $(n-1)$-form symmetry. Then we can associate the gauge field to a quadratic form $q_{\rho_n}(B)$, couple the bosonic theory with a term defined via the quadratic form, and gauge the whole theory by making $B$ dynamical. Since the quadratic term depends on a $n$-form $\rho_n$ determined only by the spacetime manifold, the gauged theory depends on $\rho_n$ as well. In the case $n=1$, this procedure is the fermionization [58], and it produces a fermionic theory from a bosonic theory. Let us continue to call this recipe "fermionization" for general $n$, even though this recipe constructs exotic phases with higher-dimensional excitations instead of fermionic particles. In this recipe, we couple the bosonic theory to the $\mathbb{Z}_2$ $n$-form gauge theory (90) that describes the invertible exotic higher topological order with $n$-group symmetry. The partition function of the new theory $F[T]$ obtained from fermionizing the theory $T$ is [37]

$$Z_{F[T]}[\rho_n] = \frac{1}{|H^n(M, \mathbb{Z}_2)|^{1/2}} \sum_{b \in H^n(M, \mathbb{Z}_2)} Z_T[b] e^{\frac{\pi i}{2} \int q_{\rho_n}(b)}, \qquad (92)$$

where $b$ is a $\mathbb{Z}_2$ $n$-form gauge field with the gauge transformation $b \to b + d\lambda$, and the left hand side depends on $\rho_n$.[38] The new theory $F[T]$ has a new $(n-1)$-form $\mathbb{Z}_2$ symmetry, which is now part of the $n$-group symmetry, since the theory depends on the background $\rho_n$ of the symmetry through the weight $e^{\frac{\pi i}{2} \int q_{\rho_n}(b)}$ in the summation (92), which is the effective action of the invertible exotic higher topological order. The background for the $(n-1)$-form symmetry is denoted by $\rho_n$. When $n=1$, the extra weight makes the theory depend on the spin structure $\rho_1$, and thus the name "fermionic", and we will continue to adopt the terminology for higher $n$. If we change the background by $\rho_n \to \rho_n + x$, then the sum changes by extra weight $(-1)^{\int b \cup x}$. We remark that when the theory $T$ has time-reversal symmetry that does not transforms the background gauge field for the $(n-1)$-form symmetry, it acts on the partition function as complex conjugation $Z_T[b]^* = Z_T[b]$, and (92) implies that the theory $F[T]$ has time-reversal symmetry that participates in $n$-group for even $n$. Note that the $n$-group for even $n$ is a nontrivial product between the Lorentz group and the $\mathbb{Z}_2$ $(n-1)$-form symmetry when the Lorentz symmetry includes time-reversal symmetry, namely when the Lorentz group is given by $O(d)$ (with $d = 2n$). When we reduce the Lorentz symmetry to $SO(d)$ by excluding the time-reversal symmetry, the $n$-group reduces to a trivial product between the Lorentz symmetry $SO(d)$ and the $\mathbb{Z}_2$ $(n-1)$-form symmetry.

The theory $F[T]$ we end up with now has a $\mathbb{Z}_2$ $(n-1)$-form symmetry whose combination with the spacetime $O(2n)$ Lorentz symmetry requires a $v_{n+1}$ Wu structure. Here, $v_{n+1}$ is the $(n+1)$th Wu class. The existence of the $v_{n+1}$ Wu structure requires $v_{n+1}$ to be exact on the $2n$-dimensional spacetime manifold $\mathcal{M}$. The $n$-form $\rho_n$ such that $v_{n+1} = d\rho_n$ specifies the $v_{n+1}$ Wu structure on $\mathcal{M}$ as discussed before.

In the following we will discuss the relation between the fermionization of $T$ and $T/\mathbb{Z}_2$ i.e. the theory obtained from $T$ by gauging non-anomalous $\mathbb{Z}_2$ $(n-1)$-form symmetry. The relation is summarized in Figure 4.

---

[36]Here, $\oint d\lambda_{b,x} = 0 \mod 2$, leaving the $n$-holonomy of $b$ and $x$ unchanged. Equivalently saying, $b, x \in H^n(M, \mathbb{Z}_2)$.

[37]We remark that while the procedure can be defined for any theory $T$, if $T$ does not have time-reversal symmetry, then the fermionization procedure in general does not produce a time-reversal invariant theory. Thus if the $n$-group symmetry requires time-reversal symmetry, we will restrict to the theories $T$ that has time-reversal symmetry.

[38]In 1+1d we can identify $q_{\rho_1}(b)/2 = \mathrm{Arf}(\rho_1 + b) - \mathrm{Arf}(\rho_1)$, and it can be compared with (3.3) in [58].

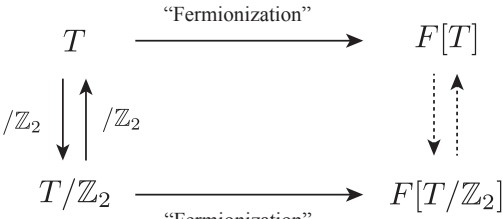

Figure 4: The interplay between fermionization and gauging $\mathbb{Z}_2$ symmetry. The downward arrow on the left denotes gauging the $\mathbb{Z}_2$ $(n-1)$-form symmetry in $T$, while the upward arrow on the left denotes gauging the dual $\mathbb{Z}_2$ $(n-1)$-form symmetry in $F[T]$. On the right, the downward arrow and upward arrow denotes stacking with the invertible phase in Section 4 or its complex conjugate, together with a change in the symmetry fractionalization as discussed in (96).

Let us compare the partition functions of $F[T]$ and $F[T/\mathbb{Z}_2]$, the latter is the "fermionization" of the bosonic theory $T/\mathbb{Z}_2$. The partition function of $F[T/\mathbb{Z}_2]$ is

$$
\begin{aligned}
Z_{F[T/\mathbb{Z}_2]}[\rho_n] &= \frac{1}{|H^n(M,\mathbb{Z}_2)|} \sum_{x,b} Z_T[b] \exp\left(\pi i \int b \cup x\right) \exp\left(\frac{\pi i}{2} \int q_{\rho_n}(x)\right) \\
&= \frac{1}{|H^n(M,\mathbb{Z}_2)|} \sum_{x,b} Z_T[b] e^{\frac{\pi i}{2} \int \left(q_{\rho_n}(x+b) - q_{\rho_n}(b)\right)} \\
&= \left(\frac{1}{|H^n(M,\mathbb{Z}_2)|^{1/2}} \sum_b Z_T[b] e^{-\frac{\pi i}{2} \int q_{\rho_n}(b)}\right) \\
&\quad \cdot \left(\frac{1}{|H^n(M,\mathbb{Z}_2)|^{1/2}} \sum_{x'} e^{\frac{\pi i}{2} \int q_{\rho_n}(x')}\right) \quad \text{with } x' = x+b \\
&= \left(\frac{1}{|H^n(M,\mathbb{Z}_2)|^{1/2}} \sum_b Z_T[b] e^{-\frac{\pi i}{2} \int q_{\rho_n}(b)}\right) Z^{\text{iETO}}[\rho_n],
\end{aligned}
\tag{93}
$$

where we used the quadratic property (B.1) to complete the square $q_{\rho_n}(x) + 2x \cup b = q_{\rho_n}(b+x) - q_{\rho_n}(b)$, and we use the change of variable $x' = b + x$. And $Z^{\text{iETO}}[\rho_n]$ is the partition function for the invertible $\mathbb{Z}_2$ $n$-form gauge theory (90).

The expression in (93) is not quite proportional to $Z_{F[T]}[\rho_n]$ since it has $-\frac{\pi i}{2} q_{\rho_n}(b)$ instead of $\frac{\pi i}{2} \int q_{\rho_n}(b)$. This can be remedied as follows: their difference is $\pi \int q_{\rho_n}(b) = \pi \int b^2 = \pi \int b \cup v_n$, so it can be expressed as $F[T]$ coupled to a background $v_n$ for the dual $\mathbb{Z}_2$ $(n-1)$ form symmetry,[39]

$$
\begin{aligned}
Z_{F[T]}[\rho_n; X] &= \frac{1}{|H^n(M,\mathbb{Z}_2)|^{1/2}} \sum_b Z_T[b] e^{\frac{\pi i}{2} \int q_{\rho_n}(b)} e^{\pi i \int b \cup X} \\
&= \frac{1}{|H^n(M,\mathbb{Z}_2)|^{1/2}} \sum_b Z_T[b] e^{\frac{\pi i}{2} \int \left(q_{\rho_n}(b+X) - q_{\rho_n}(X)\right)}, \quad X = v_n.
\end{aligned}
\tag{94}
$$

The coupling to background $v_n$ has the effect of changing the symmetry fractionlization for Lorentz symmetry on the $(n-1)$-membrane that transforms under the $(n-1)$-form symmetry: it

---

[39]This is (3.4) of [58], where on the left hand side the dependence on $X, \rho_n$ is combined into $X + \rho_n$ (so changing the spin structure $\rho_n \to \rho_n + z$ is equivalent to changing $X \to X + z$).

carries additional 't Hooft anomaly for Lorentz symmetry, as specified by $(-1)^{\nu_n} \in H^n(BO(2n), U(1))$.

Then we can rewrite the bracket in the last line of (93) as

$$\frac{1}{|H^n(M,\mathbb{Z}_2)|^{1/2}} \sum_b Z_T[b] e^{-\frac{\pi i}{2}\int q_{\rho_n}(b)} = \frac{1}{|H^n(M,\mathbb{Z}_2)|^{1/2}} \sum_b Z_T[b] e^{\frac{\pi i}{2}\int q_{\rho_n}(b) - \pi i \int b \cup b}$$

$$= \frac{1}{|H^n(M,\mathbb{Z}_2)|^{1/2}} \sum_b Z_T[b] e^{\frac{\pi i}{2}\int q_{\rho_n}(b) + \pi i \int b \cup \nu_n} = Z_{F[T]}[\rho_n; \nu_n], \tag{95}$$

where we used $\pi \int b^2 = \pi \int b \cup \nu_n \bmod 2\pi$ by Wu formula. Then we find

$$Z_{F[T/\mathbb{Z}_2]}[\rho_n] = Z_{F[T]}[\rho_n; \nu_n] Z^{\mathrm{iETO}}[\rho_n]. \tag{96}$$

In the case of 1+1d, $n = 1$, when the manifold has a spin structure, it is also orientable and so $\nu_1 = w_1$ is trivial, then the above equation implies that the fermionization of $T$ and $T/\mathbb{Z}_2$ differs simply by stacking the partition function of the Kitaev's chain invertible fermionic topological order in 1+1d. This reproduces the result in [58, 59].

Applying (96) twice leaves the theory invariant:

$$Z_{F[T/\mathbb{Z}_2/\mathbb{Z}_2]}[\rho_n] = Z_{F[T/\mathbb{Z}_2]}[\rho; \nu_n] Z^{\mathrm{iETO}}[\rho_n] = Z_{F[T/\mathbb{Z}_2]}[\rho + \nu_n] Z^{\mathrm{iETO}}[\rho_n]$$

$$= \left( Z_{F[T]}[\rho_n + \nu_n + \nu_n] Z^{\mathrm{iETO}}[\rho_n + \nu_n] \right) Z^{\mathrm{iETO}}[\rho_n] = Z_{F[T]}[\rho_n] Z^{\mathrm{iETO}}[\rho_n + \nu_n] Z^{\mathrm{iETO}}[\rho_n]$$

$$= Z_{F[T]}[\rho_n], \tag{97}$$

where we used $Z_{F[T/\mathbb{Z}]}[\rho_n; X] = Z_{F[T/\mathbb{Z}]}[\rho_n + X; 0]$ and $Z^{\mathrm{iETO}}[\rho_n + \nu_n] Z^{\mathrm{iETO}}[\rho_n] = 1$.[40]

# 6 Outlook

In this work, we present exotic invertible phases protected by higher-group symmetry that extends the spacetime Lorentz group by higher-form $\mathbb{Z}_2$ symmetries. It would be interesting to find explicit lattice models for these phases.[41] In particular, it is an intriguing question how the exotic time-reversal symmetry and the higher form symmetries that are embedded in the spacetime $n$-group can be realized in the microscopic lattice models.

It would also be interesting to explore phase transitions protected by the higher group symmetries, such as bulk phase transitions between different invertible phases as discussed in Section 3.6.4 for the transition between iELTO and the trivial phase, or boundary phase transitions protected by anomalous higher-group symmetry.

Moreover, in this work, we have considered a specific type of spacetime $n$-group which involves time-reversal symmetry and higher-form $\mathbb{Z}_2$ symmetry. The construction of the $n$-group relies on the Wu class. One direction for future research is to explore other higher group extensions of the Lorentz group. For example, one can generalize the higher-form symmetry

---

[40]The latter comes from

$$Z^{\mathrm{iETO}}[\rho_n + \nu_n] = \frac{1}{|H^2(M,\mathbb{Z}_2)|^{1/2}} \sum_b e^{\frac{\pi i}{2}\int q_{\rho_n}(b) + \pi i \int b \cup \nu_n}$$

$$= \frac{1}{|H^2(M,\mathbb{Z}_2)|^{1/2}} \sum_b e^{\frac{\pi i}{2}\int q_{\rho_n}(b) + \pi i \int b^2} = \left( \frac{1}{|H^2(M,\mathbb{Z}_2)|^{1/2}} \sum_b e^{\frac{\pi i}{2}\int q_{\rho_n}(b)} \right)^*$$

$$= \left( Z^{\mathrm{iETO}}[\rho_n] \right)^*, \tag{98}$$

where we've used $q_{\rho_n + X}(b) = q_{\rho_n}(b) + 2b \cup X$ and the Wu formula $b \cup \nu_n = b \cup b \bmod 2$.

[41]After the appearance of this work, such lattice model is constructed in [60].

part of the group to other Abelian groups. One can also change the Postnikov class that controls the mixing of the higher-form symmetry and the Lorentz symmetry in the higher spacetime group. It is interesting to study the properties of the topological phases of matter protected by such generalized higher spacetime group and find their microscopic realizations.

Another direction is to explore the SPT phases with spacetime higher-group symmetry and other internal symmetries. Just as the presence of fermion particles can change the possible symmetry protected topological phases for internal symmetry, one can also explore how the presence of exotic excitations and higher-group spacetime symmetry can modify the classifications of topological phases. For instance, there can be new phases in the presence of exotic excitations,[42] or some phases can become trivial.

# Acknowledgements

We thank Xie Chen, Anton Kapustin, Juven Wang for discussions. We thank Cenke Xu for discussions and participation in the early stage of the project. We thank Meng Cheng and Anton Kapustin for comments on a draft. The work of P.-S. H is supported by the U.S. Department of Energy, Office of Science, Office of High Energy Physics, under Award No. DE-SC0011632, and by the Simons Foundation through the Simons Investigator Award. The work of WJ is supported by NSF Grant No. DMR-1920434, the David and Lucile Packard Foundation, and the Simons Foundation.

# A  Review of higher-group symmetry

Here we summarized some properties of $n$-group symmetry that combines 0-form symmetry $K$ and $(n-1)$-form $\mathbb{Z}_2$ symmetry. For an introduction of higher-group symmetry from physics perspective, see *e.g.* [21, 30, 62, 63].

The $n$-group symmetry can be described by the extension

$$1 \to \mathbb{Z}_2 \to \mathbb{G} \to K \to 1\,. \tag{A.1}$$

The extension $\mathbb{G}$ is specified by $\omega_{n+1} \in H^{n+1}(BK, \mathbb{Z}_2)$, called the Postnikov class of the $n$-group symmetry. It is a $\mathbb{Z}_2$ valued function that depends on $(n+1)$ elements in $K$, and it satisfies the cocycle condition

$$d\omega_{n+1}(k_1, k_2, \cdots, k_{n+2}) = \sum_{i=1}^{n+2}(-1)^{i+1}\omega_{n+1}(k_1, \cdots \hat{k}_i, \cdots, k_{n+2}) = 0 \quad \mod 2\,. \tag{A.2}$$

We will explain its physical meaning and implications in the following.

**A one-group symmetry**   Let us begin with $n = 1$, where $\mathbb{G}$ is an ordinary group. The extension modifies the algebra obeyed by the symmetry generators: the action of symmetry $K$ is projective up to an action of the $\mathbb{Z}_2$ symmetry:

$$R_k R_{k'} = \omega_2(k, k')R_{kk'}\,, \tag{A.3}$$

where $R_k$ for $k \in K$ is the action of symmetry $K$, and $\omega_2(k, k')$ is the symmetry action of $\mathbb{Z}_2$ that depends on $k, k'$. In other words, if the $\mathbb{Z}_2$ symmetry acts trivially then $R$ is a linear representation of $K$, while if the $\mathbb{Z}_2$ symmetry acts non-trivially $R$ is a projective representation.

---

[42]Examples are discussed in [61].

The group cocycle $\omega_2$ determines a 1+1d invertible phase (it can be embedded in a higher dimensional space) protected by the symmetry $K$ [1,64]. In the presence of background gauge field of the group $K$ the effective action is[43]

$$\pi \int_\Sigma \omega_2. \tag{A.4}$$

On the 0+1d boundary of the invertible phase, there is a particle carrying the projective representation $(-1)^{\omega_2}$ of the group $K$. The relation (A.3) implies that such particle also transforms non-trivially under the $\mathbb{Z}_2$ symmetry.

If we turn on background gauge field $\rho_1$ for the $\mathbb{Z}_2$ symmetry and background $A$ for the $K$ symmetry, the algebra of the generators implies that

$$d\rho_1 = \omega_2(A). \tag{A.5}$$

The way to understand this relation is that the particle carries unit charge under the $\mathbb{Z}_2$ symmetry needs to attach with the Wilson line $e^{\pi i \int \rho_1}$ on the world history to be gauge invariant, and the dependence on the surface (A.4) implies the relation (A.5) by Stokes' theorem.

The relation (A.5) also implies that a background gauge transformation of $A$, which changes $\omega_2(A)$ by an exact term, also transforms the background $\rho_1$. This is related to the algebra relation (A.3). We remark that this transformation does not imply the symmetry is anomalous. The transformation of $K$ symmetry changes the background gauge field $\rho_1$ and thus produces additional symmetry defect of the $\mathbb{Z}_2$ symmetry, *i.e.* the theory is not invariant but changes by the symmetry defect, as opposite to changed by a number as in the conventional 't Hooft anomaly. In particular, this non-invariance does not need to match between the UV and the IR in the renormalization group flow.

**A two-group symmetry** For $n = 2$, where $\mathbb{G}$ is a two-group, the extension modifies the algebra obeyed by the symmetry generators: the action of symmetry $K$ has non-trivial "operator-valued" F-symbol that is the $\mathbb{Z}_2$ one-form symmetry action,

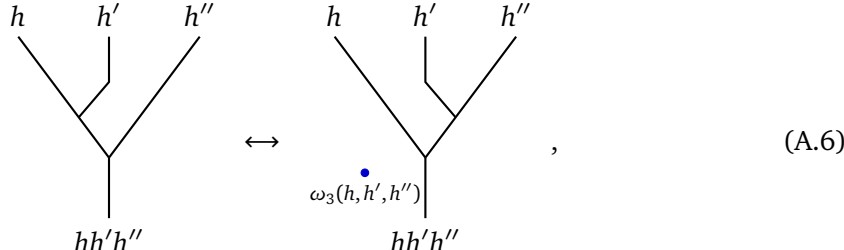

$$\tag{A.6}$$

where the blue dot is a generator of the $\mathbb{Z}_2$ one-form symmetry given by $\omega_3(h, h', h'')$.

The equation (A.6) can be phrased as the one-form object transforming under $\mathbb{G}$ in the following way,

$$R_{h,h'} R_{hh',h''} = \epsilon^{\omega_3(h,h',h'')} R_{h',h''} R_{h,h'h''}, \quad h, h', h'' \in K, \tag{A.7}$$

where $\epsilon$ is the representation of the $\mathbb{Z}_2$ one-form symmetry. A way to understand this equation is given in the discussion of (16). Another way to it is to view $R_{h,h'}$ as the action of the junction of three 0-form symmetry defects labelled by $h, h'$ and $hh'$ [65].[44] In the above equation, $\epsilon^{\omega_3(h,h',h'')} = 1, \epsilon$ for $\omega_3(h, h', h'') = 0, 1$ is an action of the $\mathbb{Z}_2$ one-form symmetry. In

---

[43]If we triangulate the surface $\Sigma$, then the effective action is the sum of $\omega_2(k_1, k_2)$ over each triangle with $K$ elements $k_1, k_2$ on the independent edges.

[44]We note that the action can be modified by projective phase, $R_{h,h'} \to R_{h,h'} \omega_2(h, h')$ for $\omega_2(h, h')$ obeys the cocycle condition $\omega_2(h, h')\omega_2(hh', h'') = \omega_2(h', h'')\omega_2(h, h'h'')$, without changing the property (A.7).

other words, for loops neutral under the $\mathbb{Z}_2$ one-form symmetry ($\epsilon = +1$), the $K$ symmetry is associative; for loops that transform non-trivially under the $\mathbb{Z}_2$ one-form symmetry ($\epsilon = -1$), the $K$ symmetry action has a non-trivial associator $(-1)^{\omega_3}$.

The group cocycle $\omega_3$, when viewed as an element in $H^3(K, U(1))$, determines a 2+1d invertible phase (embedded in a higher dimensional space)[45] protected by the symmetry $K$ [1,64]. In the presence of background gauge field of the group $K$ the effective action is

$$\pi \int \omega_3 \,. \tag{A.9}$$

On the 1+1d boundary of the invertible phase, there can be a theory that describes loops that transform non-trivially under the $\mathbb{Z}_2$ one-form symmetry. The $K$ symmetry action on the boundary has non-trivial associator $(-1)^{\omega_3}$.

If we turn on background gauge field $\rho_2$ for the $\mathbb{Z}_2$ one-form symmetry and background $A$ for the $K$ symmetry, the algebra of the generators implies that

$$d\rho_2 = \omega_3(A)\,. \tag{A.10}$$

Another way to understand this relation is to consider the loop that carries unit charge under the $\mathbb{Z}_2$ one-form symmetry. Since it transforms under $K$ as in (A.7), its world-history by itself is not invariant under the gauge transformation of $A$, instead it changes by the boundary transformation of the bulk term $e^{i\pi \int_{\mathcal{V}} \omega_3}$. Since the loop carries one-form charge, gauge-invariance under the one-form symmetry requires its world-history to attach with the Wilson surface $e^{-\pi i \int \rho_2}$ of the gauge field of $\mathbb{Z}_2$ one-form symmetry. And the dependence on the volume (A.9) implies the relation (A.10) by Stokes' theorem.

The relation (A.10) also implies that a background gauge transformation of $A$, which changes $\omega_3(A)$ by an exact term, also transforms the background $\rho_2$. This is related to the algebra relation (A.7). We remark that this transformation does not imply the symmetry is anomalous. The transformation of $K$ symmetry changes the background gauge field $\rho_2$ and thus produces additional symmetry defect of the $\mathbb{Z}_2$ one-form symmetry, *i.e.* the theory is not invariant, but changed by the symmetry defect, as opposite to changed by a number as in the conventional 't Hooft anomaly. In particular, this non-invariance does not need to match between the UV and the IR in the renormalization group flow [30].

**An $n$-group symmetry** Likewise, for general $n$, the extension is a $n$-group $\mathbb{G}$. The symmetry action has generalized F-symbol $\omega_{n+1}$ that is the leading "obstruction" to fusing $(n+1)$ elements in $K$, and it takes value in the $\mathbb{Z}_2$ one-form symmetry action. For instance, the case $n = 3$ is shown in Figure 6 of [21], with $\omega(\mathbf{g}, \mathbf{h}, \mathbf{k}, \mathbf{l})$ replaced by elements in $\mathbb{Z}_2$ one-form symmetry. We stress that the generalized F-symbol indicates a modification of symmetry algebra and does not mean the entire theory has an 't Hooft anomaly [21,30]. The extension modifies the algebra of symmetry generators This implies that the $(n-1)$-membrane that carries unit charge under the $\mathbb{Z}_2$ $(n-1)$-form symmetry carries an anomaly described by the $K$ symmetry action with the generalized F-symbol $(-1)^{\omega_{n+1}}$ (which is now a number instead of symmetry generator), and it lives on the boundary of an $(n+1)$-dimensional SPT phase with the effective action

$$\pi \int \omega_{n+1} \,. \tag{A.11}$$

---

[45]For instance, when $K$ is the $\mathbb{Z}_2$ time-reversal symmetry, the non-trivial $\omega_3 \in H^3(B\mathbb{Z}_2^T, \mathbb{Z}_2)$ corresponds to the effective action

$$\pi \int_{\mathcal{V}} w_1(TM)^3 \,, \tag{A.8}$$

where $M$ is the higher-dimensional space where the loops live, and $w_1(TM)$ is the restriction of $w_1(TM)$ to $\mathcal{V}$, which decomposes into $w_1(TM)|_{\mathcal{V}} = w_1(T\mathcal{V}) + w_1(N\mathcal{V})$ that involves both the tangent bundle of $\mathcal{V}$ and the normal bundle of $\mathcal{V}$ embedded in $M$. On the other hand, the effective action $\pi \int_{\mathcal{V}} w_1(T\mathcal{V})^3$ is trivial.

The SPT phase is specified by $(-1)^{\omega_{n+1}} \in H^{n+1}(BK, U(1))$, and if $K$ is an internal symmetry it is a group cohomology SPT phase [1, 64].

If we turn on the background gauge field $\rho_n$ for the $\mathbb{Z}_2$ $(n-1)$-form symmetry and background $A$ for the $K$ symmetry, the algebra of the generators implies that

$$d\rho_n = \omega_{n+1}(A). \tag{A.12}$$

Another way to understand this relation is that the $(n-1)$-membrane carries unit charge under the $\mathbb{Z}_2$ $(n-1)$-form symmetry needs to attach with the Wilson $n$-volume $e^{\pi i \int \rho_n}$ on the world history to be gauge invariant, and the dependence on the $(n+1)$-volume (A.11) implies the relation (A.12) by Stokes' theorem.

# B    Properties of quadratic function for intersection form

In this appendix we summarize some mathematical properties of quadratic function which refines the intersection pairing on $H^n(M, \mathbb{Z}_2)$ on $2n$-dimensional manifold $M$ [35, 66].

Denote $B$ to be $\mathbb{Z}_2$ value $n$-form gauge field. The quadratic function satisfies

$$q(B + B') = q(B) + q(B') + 2B \cup B' \mod 4, \tag{B.1}$$

where the multiplication by 2 in $2B \cup B'$ using the inclusion of $\mathbb{Z}_2$ as the even elements in $\mathbb{Z}_4$. In particular, $q(2B) = q(0) = 0 = 2q(B) + 2B^2 \mod 4$ *i.e.* $q(B) \mod 2 = B^2$.

We will use the convention

$$q(B) \mod 2 = B \cup B = B^2, \quad 2q(B) = 2B \cup B = 2B^2 \mod 4. \tag{B.2}$$

Two solutions for the quadratic function $q$ differ by a linear function $B \to 2 \int_M B \cup X$ for some $X \in H^n(M, \mathbb{Z}_2)$, so different quadratic functions on a given manifold $M$ are in one-to-one correspondence with elements in $H^n(M, \mathbb{Z}_2)$, but not in a canonical way [35].[46]

The solutions of the quadratic function are labelled by different $\rho_n$ that can appear in $d\rho_n = v_{n+1}$ where $v_{n+1}$ is the $(n+1)^{\text{th}}$ Wu class, where different $\rho_n$'s form an $H^n(M, \mathbb{Z}_2)$ torsor [35, 66] (see *e.g.* Corollary 1.17 of [35]).[47] We will denote the quadratic function by $q_{\rho_n}$. For manifold with dimension less or equal to $2n+1$, the Wu class $v_{n+1}$ is exact. Therefore, there always exists such $\rho_n$ that $d\rho_n = v_{n+1}$ [32]. Changing the Wu structure $\rho_n \to \rho_n + z_n$ by $z \in H^n(M, \mathbb{Z}_2)$ changes the quadratic function by

$$q_{\rho_n + z_n}(b) = q_{\rho_n}(b) + 2b \cup z_n. \tag{B.3}$$

The Brown-Kervaire invariant is the partition function

$$Z[\rho_n] = \frac{1}{|H^n(M, \mathbb{Z}_2)|^{1/2}} \sum_{b \in H^n(M, \mathbb{Z}_2)} e^{\frac{\pi i}{2} \int q_{\rho_n}(b)}. \tag{B.4}$$

The partition function is an eighth root of unity. It depends on the Wu structure $\rho_n$. Change the Wu structure $\rho_n \to \rho_n + z_n$ leads to the following change of the partition function:

$$Z[\rho_n]/Z[\rho_n + z] = e^{\frac{\pi i}{2} \int q_{\rho_n}(z)}. \tag{B.5}$$

---

[46]We note that the manifolds with even $n$ have canonical Pontryagin square operation $\mathcal{P}(B)$, but on unorientable manifolds it does not give a well-defined numerical quadratic function, since $e^{\frac{2\pi i}{4} \int \mathcal{P}(B)}$ is not invariant under a local orientation reversal.

[47]In the special case that $n = 1$ and on an orientable closed surface, $v_2 = w_2$. If the second Stiefel-Whitney class $w_2$ is not only exact, but also $0 \in H^2(M, \mathbb{Z}_2)$, $\rho_1$ is a 1-cocycle representing a choice of spin structure. If the second Stiefel-Whitney class $w_2$ does not vanish, $\rho_1$ is a 1-cochain representing a choice of spin structure that cannot be trivial. The trivial spin structure is that the fermion has anti-periodic (Neveu-Schwarz) boundary condition along any $S^1$ on $M$.

We remark that in the case of $n = 2$, one way to see that the $\nu_3$ Wu structure gives a well-defined quadratic function is as follows. Let us start with the Pontryagin square $\frac{\pi}{2}\mathcal{P}(B)$. It is not invariant under the time-reversal transformation but changes by $\pi B \cup B = \frac{\pi}{2}\mathcal{P}(B) - (-\frac{\pi}{2}\mathcal{P}(B))$, and thus it cannot be integrated over on unorientable manifolds. As we discussed in (80), this non-invariance can be compensated by introducing the Wu structure $\rho_2$ that couples as $\pi B \cup \rho_2$, with $d\rho_2 = \nu_3$. The quadratic function on orientable manifolds is given by the combination $\frac{\pi}{2}\int \mathcal{P}(B) + \pi \int B \cup \rho_2$, which is invariant under the time-reversal transformation, and it can be extended to unorientable manifolds. (For the correspondence between the Wu structure and the quadratic function, see [35]).

## C  Partition function of iELTO on simple manifolds

Let us compute the partition function (44) on some simple manifolds and investigate how it depends on the background $\rho_2$. To simplify the notation, in this appendix we will omit the subscript 2 that implies the background is a two-form. Since the correlation function of the worldsheet depends on $\rho_2$ as discussed in Section 3.4.2, it can also be interpreted as some kind of boundary condition for the exotic loop.

**On $S^2 \times S^2$ spacetime.**   The spacetime is $S^2$ in $t, x$ direction, and $S^2$ in the $y, z$ direction. There are four possible values of background $\rho$, specified by its value on different spheres:

$$(\rho_{01}, \rho_{23}), \quad \rho_{01}, \rho_{23} = 0, 1. \tag{C.1}$$

The action (34) evaluates to (omitting an overall factor $\pi/2$)

$$\int_{S^2 \times S^2} q(B) = 2\rho_{01}B(S^2_{01}) + 2\rho_{23}B(S^2_{23}) + 2B(S^2_{01})B(S^2_{23}), \tag{C.2}$$

where $B(S^2_{ij})$ is two-holonomy in the $i$th and $j$th dimensions. Since different $B$ corresponds different holonomies on the two closed surfaces, summing over possible $B$ is the same as summing over the possible holonomies on $S^2 \times S^2$ spacetime. Therefore, the partition function is

$$Z^{\text{iELTO}}[\rho_{01}, \rho_{23}] = \frac{1}{|H^2(S^2 \times S^2, \mathbb{Z}_2)|^{1/2}} \sum_{B(S^2_{ij})=0,1} e^{i\frac{\pi}{2}\int_{S^2 \times S^2} q(B)}$$

$$= \frac{1}{2}\left[1 + (-1)^{\rho_{01}} + (-1)^{\rho_{23}} - (-1)^{\rho_{01}+\rho_{23}}\right], \tag{C.3}$$

where we used $|H^2(S^2 \times S^2, \mathbb{Z}_2)| = 2^2$. It is 1 for $(\rho_{01}, \rho_{23}) = (0,0), (0,1), (1,0)$ and $-1$ for $(\rho_{01}, \rho_{23}) = (1,1)$.[48]

We can ask how are the partition functions with different background $\rho$ are related by $SL(2, \mathbb{Z})$ transformation on $S^2 \times S^2$ [49]. The transformation on $S^2 \times S^2$ is the analogue of the modular group of $S^1 \times S^1$.

Consider the transformation, $\tau \to \tau + 1$, $S^2_{23}$ is wrapped over once more by $S^2_{01}$.

$$Z^{\text{iELTO}}[0, 1] \to Z^{\text{iELTO}}[1, 1] = -Z^{\text{iELTO}}[0, 1]. \tag{C.4}$$

---

[48]We remark that this is similar to the partition function of 1+1d Kitaev's chain fermionic invertible topological order on $S^1 \times S^1$, where the partition function equals $-1$ for odd spin structure and $+1$ for even spin structure.

[49]The transformation associated with $\begin{pmatrix} a & b \\ c & d \end{pmatrix} \in SL(2, \mathbb{Z})$ is given by mapping the first $S^2$ to $S^2 \times S^2$ with degree $(a, b)$ (where the degree of the map $S^2 \to S^2$ equals the number of times the sphere is wrapped) and the second $S^2$ to $S^2 \times S^2$ with degree $(c, d)$. For general $a, b, c, d$ it is not a diffeomorphism, since it does not preserve the intersection form. The true mapping class group is the diheral group of order 8 (see e.g. [67]). The same applies to the modular group action on $S^1 \times S^1$, where $SL(2, \mathbb{Z})$ is the mapping class group.

**On $S^1 \times S^1 \times S^2$ spacetime.** The spacetime is $S^1$ in each of the 0 and 1 directions, and $S^2$ in the 2, 3 directions. There are four components of the background $\rho$, specified by

$$\rho = (\rho_{01}, \rho_{23}), \quad \rho_{01}, \rho_{23} = 0, 1. \tag{C.5}$$

The action is

$$\int_{S^1 \times S^1 \times S^2} q(B) = 2\rho_{01} B(S^1 \times S^1) + 2\rho_{23} B(S^2) + 2B(S^1 \times S^1)B(S^2), \tag{C.6}$$

where $B(S^1 \times S^1)$ and $B(S^2)$ are 2-holonomy in $S_0^1 \times S_1^1$ and $S_{23}^2$, respectively. Since different $B$ corresponds different holonomies on the two closed surfaces, summing over possible $B$ is the same as summing over the possible holonomies. Therefore, the partition function is

$$Z^{\text{iELTO}} = \frac{1}{|H^2(S^1 \times S^1 \times S^2, \mathbb{Z}_2)|^{1/2}} \sum_{B(S_i^1)=0,1} e^{i\frac{\pi}{2} \int_{S^1 \times S^1 \times S^2} q(B)}$$

$$= \frac{1}{2} \left[ 1 + (-1)^{\rho_{01}} + (-1)^{\rho_{23}} - (-1)^{\rho_{01}+\rho_{23}} \right], \tag{C.7}$$

where we used $|H^2(S^1 \times S^1 \times S^2, \mathbb{Z}_2)| = 2^2$. It is 1 for $(\rho_{01}, \rho_{23}) = (0,0), (0,1), (1,0)$ and $-1$ for $(\rho_{01}, \rho_{23}) = (1,1)$.

**On $T^4 = S^1 \times S^1 \times S^1 \times S^1$ spacetime.** There are $2^6$ kinds of boundary conditions, specified by possible $\mathbb{Z}_2$ 2-holonomy for each torus on $T^4$, there are $\binom{4}{2} = 6$ of them, labelled by $\rho_{ij}$ with $0 \le i < j \le 3$.

To compute the partition function, we need to specify the quadratic function. Let us first find a quadratic function on $T^4 = S_0^1 \times S_1^1 \times S_2^1 \times S_3^1$. The intersection happens between $S_i^1 \times S_j^1$ and $S_k^1 \times S_l^1$ with distinct $i, j, k, l$. A quadratic function for zero background with $\rho_{ij} = 0$ for all $\rho_{ij}$ is given by[50]

$$q_0(B)(T^4) = 2B(S_0^1 \times S_1^1)B(S_2^1 \times S_3^1)$$
$$+ 2B(S_0^1 \times S_2^1)B(S_1^1 \times S_3^1) + 2B(S_0^1 \times S_3^1)B(S_1^1 \times S_2^1). \tag{C.9}$$

For a general background $\rho_{ij}$, the quadratic function is[51]

$$q_\rho(B)(T^4) = q_0(B)(T^4) + 2\sum_{i<j} \rho_{ij} B(S_i^1 \times S_j^1). \tag{C.10}$$

The partition function for the quadractic function action $\frac{\pi}{2} \int q_\rho(B)$ with background $\rho$ that only has nonzero $\rho_{01}, \rho_{23}$ components, equals

$$Z[\rho] = \frac{1}{2} \left[ 1 + (-1)^{\rho_{01}} + (-1)^{\rho_{23}} - (-1)^{\rho_{01}+\rho_{23}} \right], \tag{C.11}$$

where $\rho_{ij}$ is the holonomy of the background $\rho$ on $S_i^1 \times S_j^1$. The above partition function only depends on $\rho$ through $\rho' = \rho_{01}\rho_{23} \in \mathbb{Z}_2$, and it equals $Z = (-1)^{\rho'}$.

---

[50]One can verify it satisfies (B.1):

$$q_0(B + B') - q_0(B) - q_0(B')$$
$$= 2\left[ B(S_0^1 \times S_1^1)B'(S_2^1 \times S_3^1) + B(S_0^1 \times S_2^1)B'(S_1^1 \times S_3^1) + B(S_0^1 \times S_3^1)B'(S_1^1 \times S_2^1) \right]$$
$$+ 2\left[ B'(S_0^1 \times S_1^1)B(S_2^1 \times S_3^1) + B'(S_0^1 \times S_2^1)B(S_1^1 \times S_3^1) + B'(S_0^1 \times S_3^1)B(S_1^1 \times S_2^1) \right]. \tag{C.8}$$

[51]The factor 2 in front of $2\rho B$ ensures that $q(B) = q(B + 2) \mod 4$, since $B$ is $\mathbb{Z}_2$ valued, while $q(B)$ is $\mathbb{Z}_4$ valued.

# D   Continuum description of iELTO

In this appendix we discuss the continuum description of the iELTO phase using $U(1)$ gauge fields, and apply the method to discuss states on the surface of iELTO phase.

## D.1   Continuum description for the iELTO 2-form gauge theory

On orientable manifolds, the iELTO phase can be described by $U(1)$ two-form gauge field $b$ and one-form $U(1)$ gauge field $a$ with the action [13, 39]

$$\frac{2}{4\pi}bb + \frac{2}{2\pi}bda\,. \tag{D.1}$$

In the absence of boundary, the action is invariant under the one-form gauge transformation

$$b \to b + d\lambda, \quad a \to a - \lambda\,. \tag{D.2}$$

After integrating out the gauge field $a$, we are left with two-form gauge theory with action $\frac{2}{4\pi}bb$ with the gauge field $b$ constrained to have $\mathbb{Z}_2$ holonomy $\oint b \in \pi\mathbb{Z}$, and we recover the iELTO theory with quadratic action for the two-form $\mathbb{Z}_2$ gauge field without background gauge field $\rho_2$.

## D.2   Correlation function of loop excitation

Here we compute the correlation function of the loop excitation on $S^4$ in the continuum notation. We insert loop operator $\oint_\gamma a + \int_\Sigma b$ where $\partial\Sigma = \gamma$,

$$\oint_\gamma a + \int_\Sigma b + \int \frac{1}{2\pi}bb + \frac{2}{2\pi}bda = \int a\delta(\gamma)^\perp + b\delta(\Sigma)^\perp + \int \frac{1}{2\pi}bb + \frac{2}{2\pi}bda\,, \tag{D.3}$$

where $\delta(\gamma)^\perp$ is the delta function three-form that restricts the integral to $\gamma$, and similar for $\delta(\Sigma)^\perp$. Integrating out $a$ gives

$$db = -\pi\delta(\gamma)^\perp\,, \tag{D.4}$$

which can be solved in $S^4$ (or more generally, for homologically trivial $\gamma$) as

$$b = -\pi\delta(\Sigma)^\perp\,. \tag{D.5}$$

Thus the correlation function is

$$\langle\exp(\oint_\gamma a + \int_\Sigma b)\rangle = \exp\left(-\frac{\pi i}{2}\int \delta(\Sigma)^\perp\delta(\Sigma)^\perp\right)\,. \tag{D.6}$$

In particular, if we cut open the manifold, the loop on the boundary describes an anti-semion of topological spin $-i$.

We remark that the correlation function requires a framing.

**Figure-8 experiment**   One way to see the loop describes an anti-semion on the boundary is using the following process. It represents an open surface in spacetime.

One way to argue that the anti-semion gives the correct exotic loop statistics is as follows.[52] The exotic loop requires a framing and, hence can represented as a closed ribbon in the 3 spatial

---

[52]We are grateful to Cenke Xu and Xie Chen to help us make this explicit connection.

dimensions. Take a closed ribbon and and make it self-intersect in the transverse direction (passing through the intersecting point) as the shape of a "figure 8", depicted as follows:

Two ribbons differed by a $4\pi$ twist: 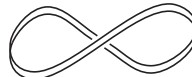 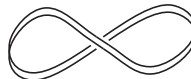

.

Then we can unwind the ribbon by $4\pi$ to recover the original ribbon. For an exotic loop this produces a minus sign: in $3+1$d spacetime, the surface operator interpolating the two "figure 8" has a single intersection point, at which the "figure 8" passing through itself in the transverse direction. This then implies that the ribbon is associated with a spin $\pm\frac{1}{4}$ particle, in agreement with the anti-semion on the boundary of the iELTO. This is consistent with the correlation function for open surfaces in Figure 7 of [29].

We remark that the same minus sign also appears in the ground state wavefunction of the semion Walker-Wang model in [68] and the model for the $m=1$ one-form $\mathbb{Z}_2$ symmetry SPT phase in [38]. These models do not have time-reversal symmetry, and this suggests upon breaking the time-reversal symmetry in the iELTO phase it can be connected to the phases described by these models. Indeed, the partition function of iELTO phase on orientable manifold is given by

$$\exp\left(\frac{i}{96\pi}\int \mathrm{Tr}\, R\wedge R - \frac{2\pi i}{4}\int \mathcal{P}(\rho_2)\right). \tag{D.7}$$

Breaking the time-reversal symmetry allows us to separately define the two terms. In particular, without time-reversal symmetry one can continuously tune the coefficient of the $\mathrm{Tr}\, R\wedge R$ term to zero and the partition function reduces to that of the one-form symmetry SPT phase. On orientable manifolds, the partition function is the same as the Walker-Wang model.

# E  "Fermionizations" of $\mathbb{Z}_2$ one-form SPTs.

In this appendix, we "fermionize" the $\mathbb{Z}_2$ one-from SPTs by working out the partition functions. Consider the partition function of SPT phase with one-form symmetry [37,69]

$$Z_T[B] = e^{\frac{m\pi i}{2}\int \mathcal{P}(B)}, \tag{E.1}$$

where $m=0,2$ preserve time-reversal symmetry while $m=1,3$ are related by time-reversal symmetry. $B$ is the background $\mathbb{Z}_2$ 2-form gauge field associated to the $\mathbb{Z}_2$ one-from symmetry.

We can gauge the $\mathbb{Z}_2$ one-form symmetry and obtain the partition function

$$Z_{T/\mathbb{Z}_2}[B'] = \sum_b e^{\frac{m\pi i}{2}\int \mathcal{P}(b)} e^{\pi i \int b\cup B'}, \tag{E.2}$$

where $B'$ is the background $\mathbb{Z}_2$ 2-form gauge field associated to the dual $\mathbb{Z}_2$ one-from symmetry.

Now we compute the partition functions of the fermionizations of the bosonic theories $T$ and $T/\mathbb{Z}_2$, begining with the time reversal symmetric cases.

## E.1  $m=0,2$: bosonic time-reversal symmetric theories

We have summarized the results in Table 2. The details are as follows,

Table 2: $\mathbb{Z}_2$ one-form SPT phases and the theories obtained by gauging the one-form symmetry and "fermionizations". Here, $v_2$ is the second Wu class. In the table, the partition functions $\delta_{B',0}, \delta_{B'+v_2,0}$ indicate a spontaneously broken one-form symmetry. In fact, the two theories are the $\mathbb{Z}_2$ gauge theories with bosonic particle and fermionic particle, respectively. We note that, the latter partition function indicates the theory has additional symmetry fractionalziation compared to the former theory, because the background $B'$ is shifted: in the latter theory, the particle created by the line operator with one-form symmetry charge is a fermion and a Kramers singlet, which is the projective representation specified by $(-1)^{v_2} \in H^2(BO(4), U(1))$ [26].

| $m$ | $Z_T[B]$ | $Z_{T/\mathbb{Z}_2}[B']$ | $Z_{F[T]}[\rho_2]$ | $Z_{F[T/\mathbb{Z}_2]}[\rho_2]$ |
|---|---|---|---|---|
| 0 | 1 | $\delta_{B',0}$ | $Z^{\text{iELTO}}[\rho]$ | 1 |
| 2 | $e^{i\pi \int B^2}$ | $\delta_{B',v_2}$ | $Z^{\text{iELTO}}[\rho_2]^*$ | $Z^{\text{iELTO}}[\rho_2]^2$ |

**Case $m = 0$** Consider $T$ to be $\mathbb{Z}_2^{(1)}$ SPT with $m = 0$, the partition function is

$$Z_T[B] = 1. \tag{E.3}$$

The theory $T/\mathbb{Z}_2$ has the partition function

$$Z_{T/\mathbb{Z}_2}[B'] = \frac{1}{|H^2(M, \mathbb{Z}_2)|^{1/2}} \sum_b e^{\pi i \int b \cup B'} = \delta_{B',0}. \tag{E.4}$$

The fermionic cousins are the following ones,

$$Z_{F[T]}[\rho_2] = \frac{1}{|H^2(M, \mathbb{Z}_2)|^{1/2}} \sum_b e^{i\frac{\pi}{2} \int q_{\rho_2}(b)} = Z^{\text{iELTO}}[\rho_2],$$

$$Z_{F[T/\mathbb{Z}_2]}[\rho_2] = \frac{1}{|H^2(M, \mathbb{Z}_2)|} \sum_b e^{i\frac{\pi}{2} \int [q_{\rho_2}(b) + 2b \cup v_2 + q_{\rho_2}(b)]}, = \frac{1}{|H^2(M, \mathbb{Z}_2)|} \sum_b e^{i\pi \int q_{\rho_2}(b) + b \cup v_2}$$

$$= \frac{1}{|H^2(M, \mathbb{Z}_2)|} \sum_b e^{i2\pi \int b \cup v_2} = 1, \tag{E.5}$$

where we have used that $e^{i\pi \int q_{\rho_2}(b)} = e^{i\pi \int b^2} = e^{i\pi \int b \cup v_2}$.

**Case $m = 2$** Consider the one-form symmetry SPT phase with the partition function

$$Z_T[B] = e^{\pi i \int \mathcal{P}(B)} = e^{\pi i \int B^2}, \tag{E.6}$$

where we used $\mathcal{P}(B) \bmod 2 = B^2 \bmod 2 = B \cup B$. It corresponds to $N = 2, p = 2$ in [37] and $n = 2, m = 2$ in [38]. It happens that because of the Wu formula $B^2 = B \cup v_2 \pmod 2$, the partition function is the same as

$$Z_T[B] = e^{\pi i \int B \cup v_2}. \tag{E.7}$$

The theory $T/\mathbb{Z}_2$ has the partition function

$$Z_{T/\mathbb{Z}_2}[B'] = \sum_b e^{\pi i \int b^2 + b \cup B'} = \sum_b e^{\pi i \int b \cup (v_2 + B')} = \delta_{B',v_2}. \tag{E.8}$$

What are their fermionic cousins? The partition function of $F[T]$ is

$$Z_{F[T]}[\rho_2] = \sum_b e^{\pi i \int b^2 + \frac{\pi i}{2} \int q_{\rho_2}(b)} = \left( \sum_b e^{-\frac{\pi i}{2} \int q_{\rho_2}(b)} \right)^* = Z^{\text{iELTO}}[\rho_2]^*, \tag{E.9}$$

where we used $q_{\rho_2}(b + v_2) - q_{\rho_2}(v_2) = q_{\rho_2}(b) + 2b \cup v_2$. The fermionic cousin $F[T]$ is the time reversal partner of iELTO.

Similarly, the partition function of $F[T/\mathbb{Z}_2]$ is

$$Z_{F[T/\mathbb{Z}_2]}[x + \rho_2] = \sum_{b'} \delta_{b', v_2} e^{\pi i \int b' \cup x} e^{\frac{\pi i}{2} q_{\rho_2}(b')} = e^{\frac{\pi i}{2} q_{\rho_2}(v_2) + \pi i \int x \cup v_2},$$

$$Z_{F[T/\mathbb{Z}_2]}[\rho_2] = Z_{F[T]}[\rho_2 + v_2] Z^{\text{iELTO}}[\rho_2] = Z^{\text{iELTO}}[\rho_2 + v_2]^* Z^{\text{iELTO}}[\rho_2] = (Z^{\text{iELTO}}[\rho_2])^2.$$

(E.10)

## E.2 $m = 1, 3$

These theories are not time-reversal symmetric and they are not in the class of bosonic theories that we are considering. Nevertheless, we can formally perform the fermionization operation. We summarize their partition functions together with those of their fermionizizations in Table 3. With only $\mathbb{Z}_2$ one-form symmetry, the $\mathbb{Z}_2^{(1)}$ orbifold of the $m$ class $\mathbb{Z}_2^{(1)}$ non-trivial SPT phase is the same phase as the $m$ class SPT phase. As shown in Table 3, the phase factor $e^{i\theta\sigma}$ that differs the partition function of the two theories can be continuously changed from 1 to $e^{i\frac{\pi}{4}\sigma}$, at the price of breaking time reversal symmetry. Furthermore, we can see from the partition functions that $m = 1$ and $m = 3$ bosonic SPT phases are time-reversal partners. The table is

Table 3: The $\mathbb{Z}_2$ one-form orbifold of $\mathbb{Z}_2$ one-form SPTs and their fermionizations. Here, $v_2$ is the second Wu class.

| $m$ | $Z_T[B]$ | $Z_{T/\mathbb{Z}_2}[B']$ | $Z_{F[T]}[\rho]$ | $Z_{F[T/\mathbb{Z}_2]}[\rho]$ |
|---|---|---|---|---|
| 1 | $e^{i\frac{\pi}{2}\int B^2}$ | $e^{i\pi\frac{\sigma}{4}} e^{-i\frac{\pi}{2}\int B'^2}$ | $\delta_{\rho_2, v_2}$ | $\delta_{\rho_2, 0} Z^{\text{iELTO}}[\rho_2]$ |
| 3 | $e^{i\frac{3\pi}{2}\int B^2}$ | $e^{-i\pi\frac{\sigma}{4}} e^{i\frac{\pi}{2}\int B'^2}$ | $\delta_{\rho_2, 0}$ | $\delta_{\rho_2, v_2} Z^{\text{iELTO}}[\rho_2]$ |

obtained from the following computations.

**Case $m = 1$** The partition function for $T/\mathbb{Z}_2$ is

$$Z_{T/\mathbb{Z}_2}[B'] = \frac{1}{|H^2(M, \mathbb{Z}_2)|^{1/2}} \sum_b e^{\frac{\pi i}{2} \int q_0(b) + \pi i \int b \cup B'}$$

$$= \frac{1}{|H^2(M, \mathbb{Z}_2)|^{1/2}} \sum_b e^{\frac{\pi i}{2} \int q_0(b + B') - \frac{\pi i}{2} \int q_0(B')} = Z^{\text{iELTO}}[0] e^{-\frac{\pi i}{2} \int q_0(B')}. \quad (E.11)$$

The partition function for $F[T]$ is

$$Z_{F[T]}[\rho_2] = \sum_b e^{i\frac{\pi}{2} \int \left(-3q_0(b) + q_{\rho_2}(b)\right)}$$

$$= \sum_b e^{\frac{\pi i}{2} \int (-2q_0(b) + 2b \cup \rho_2)} = \sum_b e^{\pi i \int (q_0(b) + b \cup \rho_2)} = \sum_b e^{\pi i \int (b^2 + b \cup \rho_2)}$$

$$= \delta_{\rho_2, v_2}. \quad (E.12)$$

The partition function for $F[T/\mathbb{Z}]$ is

$$Z_{F[T/\mathbb{Z}_2]}[\rho_2] = \sum_{b, b'} e^{\frac{\pi i}{2} \int q_0(b) + \pi i \int b \cup b' + \frac{\pi i}{2} \int q_{\rho_2}(b')} \stackrel{b'' = b + b'}{=\!=\!=} \left( \sum_{b''} e^{\frac{\pi i}{2} \int q_0(b'')} \right) \sum_{b'} e^{\pi i \int b' \cup \rho_2},$$

$$= \left( Z^{\text{iETO}}[0] \right) \sum_{b'} e^{\pi i \int b' \cup \rho_2} = \left( Z^{\text{iETO}}[0] \right) \delta_{\rho_2, 0}. \quad (E.13)$$

**Case** $m = 3$    The partition function for $T/\mathbb{Z}_2$ is

$$Z_{T/\mathbb{Z}_2}[B'] = \frac{1}{|H^2(M,\mathbb{Z}_2)|^{1/2}} \sum_b e^{-\frac{\pi i}{2}\int q_0(b)+\pi i \int b \cup B'}$$

$$= \frac{1}{|H^2(M,\mathbb{Z}_2)|^{1/2}} \sum_b e^{-\frac{\pi i}{2}\int q_0(b+B')+\frac{\pi i}{2}\int q_0(B')} = Z^{\text{iELTO}}[0]^* e^{\frac{\pi i}{2}\int q_0(B')}. \quad \text{(E.14)}$$

$$Z_T[B] = e^{-i\frac{\pi}{2}\int \mathcal{P}[B]}, \qquad\qquad Z_{T/\mathbb{Z}_2}[B] = e^{i\pi\frac{\sigma}{4}} e^{-i\frac{\pi}{2}\int \mathcal{P}[B]},$$

$$Z_{F[T]}[\rho_2] = \delta_{\rho_2,0}, \qquad\qquad Z_{F[T/\mathbb{Z}_2]}[\rho_2] = \delta_{\rho_2,0} e^{i\pi\frac{\sigma}{4}}. \qquad \text{(E.15)}$$

The derivation is as follows.

$$Z_T[B] = e^{-i\frac{2\pi}{4}\int \mathcal{P}[B]},$$

$$Z_{T/\mathbb{Z}_2}[x] = \sum_b e^{-i\frac{\pi}{2}\int[\mathcal{P}[b]-2b\cup x]} = e^{-i\frac{\theta_g}{48}\int \frac{\text{Tr}R\wedge R}{(2\pi)^2}} e^{-i\frac{2\pi}{4}\int \mathcal{P}[x]} e^{i\pi\int x\cup x}$$

$$= \left(\sum_b e^{-\frac{\pi i}{2}\int q_0(b)}\right) e^{\frac{\pi i}{2}\int q_0(x)}, \qquad\qquad \text{(E.16)}$$

where we have used that

$$\sum_b e^{-i\frac{\pi}{2}\int[\mathcal{P}[b]-2b\cup x]} = \sum_b e^{-i\frac{\pi}{2}\int[q_0(b)-2b\cup x]} = \sum_b e^{-i\frac{\pi}{2}\int[q_0(b+x)-2b\cup x-2x\cup x]}$$

$$= \sum_b e^{-i\frac{\pi}{2}\int[q_0(b)+q_0(x)-2x\cup x]}. \qquad\qquad \text{(E.17)}$$

The fermionized theory is

$$Z_{F[T]}[\rho_2] = \sum_b e^{i\frac{\pi}{2}\int\left(3q_0(b)+q_{\rho_2}(b)\right)} = \sum_b e^{\frac{\pi i}{2}\int(4q_0(b)+2b\cup\rho_2)} = \sum_b e^{\pi i \int b\cup\rho_2} = \delta_{\rho_2,0}, \quad \text{(E.18)}$$

where in the second equality we used $q_{\rho_2}(b) = q_0(b) + 2b \cup \rho_2$.

Similarly, the partition function for $F[T/\mathbb{Z}_2]$ is

$$Z_{F[T/\mathbb{Z}_2]}[\rho_2] = \sum_{b,b'} e^{-\frac{\pi i}{2}\int q_0(b)+\pi i \int b\cup b'+\frac{\pi i}{2}\int q_{\rho_2}(b')} = \sum_{b''=b-b'} e^{-\frac{\pi i}{2}q_0(b'')} \sum_{b'} e^{\pi i \int b'\cup b'+\pi i \int b'\cup\rho_2}$$

$$= Z^{\text{iELTO}}[0]^* \sum_{b'} e^{\pi i \int b'\cup b'+\pi i \int b'\cup\rho_2} = Z^{\text{iELTO}}[0]^* \sum_{b'} e^{\pi i \int b'\cup(-v_2+\rho_2)} = Z^{\text{iELTO}}[0]^*\delta_{\rho_2,v_2},$$

$$\text{(E.19)}$$

where the 4th equality used the Wu formula $b' \cup b' + b' \cup v_2 = 0$.

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
