# Peer review of "Exotic Invertible Phases with Higher-Group Symmetries"

_SciPost Physics, doi:SciPost Phys. 12, 052 (2022)_

## Round 1 · Referee Report · Anonymous (Referee 1) · 2021-11-16

Strengths

1) This is an impressive work where the authors construct and discuss many properties of a new class of invertible topological phase matter protected by a higher-form symmetry, which combines the full Lorentz group $O(d)$ with a one-form $\mathbb{Z}_2$ symmetry in a nontrivial way.

2) The resulting phases possess a number of quite interesting properties such as loop excitation that carries projective representation of the Lorentz symmetry, and boundary states with anti-semion statistics and nonvanishing thermal Hall conductance, in contrast to the usual time-reversal strictly 2+1d bosonic systems.

3) The authors discuss many consequences and connections with other topological phases.

Weaknesses

1) I find the motivations not well justified from the physical point of view. In spite of the fact that this construction generalizes the celebrated Kitaev chain, it seems to be hard to envision a physics-based model (for example, an explicit lattice model) realizing such exotic properties. Perhaps, the authors could try to provide a more compelling motivation for the phases studied in this paper.

2) This is a difficult work to read because it is very technical and lengthy, which make it not much accessible for a broader audience of physics community.

3) The excessive number of footnotes makes the reading intermittent.

Report

This is a very technical paper, where the authors study a class of invertible topological phases protected by a higher-form symmetry. The higher form symmetry is obtained through the extension of the Lorentz group by a $\mathbb{Z}_2$ one-form symmetry. As these symmetries do not commute between themselves, this extension leads to a two-group. The results are interesting and the paper discuss many consequences of the higher-form and explore many connections with other topological phases. The results are sound, timely, and merit publication in SciPost. However, prior to recommend the publication, I would suggest the authors to address some questions and comments below.

Requested changes

Questions and Comments

1) In the case of the Kitaev chain, it is possible to understand that the groundstate is unique from the action (2.6)? How to see that this theory is gapped? Here, I also suggest the authors to provide a more transparent connection with the physics of the Kitaev chain. As this is a very concrete lattice model, this would facilitate the understanding of the spirit of the remaining sections of the manuscript (related to this, see the question below).

2) Is the gauging procedure of the Kitaev chain equivalent to the one presented in SciPost Phys. 10, 148 (2021), "Gauging the Kitaev chain", by U. Borla, R. Verresen, J. Shah, and S. Moroz? Please, it would be helpful to clarify this point.

3) The action (3.21) describes the 3+1d invertible phase protected by the higher-form symmetry. As in my first question, how to see that it is gapped and that the groundstate in unique?

4) In Sec. 3.3, the authors discuss that the theory described by the action (3.21) has a $\mathbb{Z}_8$ classification, but to argue this they need to consider many copies of the theory. Is this procedure necessary? For example, this can be derived from the continuum Lagrangian presented in (D.1)?

5) I am confused with the meaning of the chiral central charge discussed in Sec. 3.5.2 and the interpretation of the corresponding 1+1d CFT, since the theory is 3+1d. This CFT can be derived from the action (3.21), or from the 2+1 dimensional action (3.48)? Perhaps, the authors could clarify this point.

6) What is the meaning of $m=3$ in the first paragraph of Sec. 3.6? It is not defined at that point.

7) Please, check the sign of the term $\lambda_{12}$ in the action (3.68).

8) In first bullet in pg 34, case $M^2<0$, the authors mean $\lambda'$ instead of $\lambda_{12}$, right?

9) In Fig. 3, the authors present the phase diagram of the Kitaev chain. What is the analogous action of (3.68) for the Kitaev chain in terms of the fermion mass $m$?

Typos: I have found many typos along the manuscript. I suggest the authors to make a careful revision. Here are some of them:

Main text

1) Right below Eq. (2.1), the word "class" is duplicate;

2) Above Eq. (2.4), the bulk theory should be in $d+1$, right?

3) In the first sentence of Sec. 3.1: "an detailed" should be "a detailed";

4) Six lines below Eq. (3.3): "...acing..."should be "acting";

5) In the second line of Sec. 3.1.2: .... "on it boundary";

6) In the fist line of pg 14: "... energy scale. that only emerges ...";

7) In the penultimate line of Sec. 3.1.4: "... namly...";

8) In the line right below Eq. (3.19): "...that charged..." should be ... that is charged;

9) In the paragraph below Eq. (3.22) the authors make reference in the future to previous section. Please, check this.

10) In the first line of the second paragraph below Eq. (3.22): "... denote..." should be ... denoted...;

11) In the first line of Sec. 3.3: "Let us show the invertible..." should be Let us show that the invertible...;

12) In the line right below Eq. (3.40) there is a duplicate comma;

13) In the paragraph below Eq. (3.53), there is a duplicate word "the";

Appendix

1) In the first sentence of Appendix A, "0-from" should be "0-form";

2) In Appendix A, in the sentence below eq. (A.1), "$\mathbb{Z}_2$ value" should be "$\mathbb{Z}_2$ valued";

3) In Appendix A, right Eq. (A.7), there is an unnecessary "is" in the sentence: "A way to understand {\it is} this equation ..."

4) In Appendix B, right above Eq. B.1, "the" should be "The";

In Appendix B, right below Eq. B.4, "toot" should be "root" ;

  • validity: high
  • significance: high
  • originality: high
  • clarity: good
  • formatting: good
  • grammar: good

Author:  Po-Shen Hsin  on 2021-11-23  [id 1969]

(in reply to Report 1 on 2021-11-16)
Category:
answer to question

Thanks for the questions and pointing out the typos, which will be fixed in v2.

1) One way to see this is by stacking the complex conjugate of the theory and show that the resulting product theory is trivial. Thus the theory (2.6) is invertible, and this can only happen if the theory is gapped with a unique ground state. Explicitly, denote the gauge field in the complex-conjugated theory by b', pi/2 \int q_{rho_2}(b) - pi/2 \int q_{rho_2}(b')
= pi/2 \int q_{rho}(b'') -\pi \int b'' cup b', b'':=b+b' where we used the property of the quadratic function q(b+b')= q(b)+q(b')+2 b cup b', 2q(b')=b' cup b'. Thus b' acts as Lagrangian multiplier that forces b''=0 and the product theory is trivial, and (2.6) is invertible i.e. gapped with a unique ground state.

Another way to see the theory is gapped with a unique ground state is using the equation of motion of b under variation b-> b+Delta b and the property of quadratic function q \pi/2\int q(b+Delta) - pi/2\int q(b) = \pi\int b\cup \Delta b + pi/2 q(Delta b) where rho_2 is omitted. Thus the equation of motion from the linear term in Delta b forces b to be trivial and the theory is gapped with a unique vacuum.

The above manipulations only used the property of the quadratic function q and apply also to the generalizations (3.21),(4,4) in higher spacetime dimensions.

2) They are different. The paper mentioned in the report discussed gauging the fermion parity symmetry, and thus the theory becomes bosonic. In contrary, in (2.6) we did not gauge the fermion parity symmetry and the theory is fermionic. The theory (2.6) can be interpreted as gauging the internal Z_2 symmetry in the SPT phase with Z_2 x Z_2^f symmetry whose effective action is (2.6), with b being the dynamical gauge field for the Z_2 internal symmetry and rho_1 being the background gauge field for the fermion parity Z_2^f symmetry.

3) see 1)

4) To study the classification of the invertible phase using the partition function, one needs to include the most general background. (E.g. to see the Z16 classification of 3+1d fermionic SPT with T^2=(-1)^F using the partition function, one needs to consider unorientable Pin+ manifold.) The continuum Lagrangian (D.1) is on orientable manifold and zero background rho_2=0. When nonzero background rho_2 is included, the partition function (D.7) on orientable manifold is an 8 root of unity. However, the symmetry group of the theory includes the time-reversal symmetry, so it is important to study the theory also on unorientable manifold. This is why in section 3.3 we need to study the classification by taking copies, where the discussion also applies on unorientable manifolds.

5) For example, a boundary of the 3+1d theory is the 2+1d U(1)_{-2} Chern-Simons theory with anti-semion, and the chiral central charge c is the framing anomaly of this topological quantum field theory. If the 2+1d Chern-Simons theory itself were to have an edge, then c would also be the chiral central charge of the edge mode. We will clarify this in v2.

6) The parameter m=0,1,2,3 labels the four different SPT phases with Z_2 one-form symmetry, with partition function (E.1). We will add the explanation and a pointer to (E.1) in v2.

7) We will change the sign in v2, and also add that the magnitude of lambda_{12} is much smaller than lambda_i and lambda'.

8) yes it should be lambda'. Thanks for spotting the typo.

9) The action in 1+1d similar to (3.68) can be Z_2 gauge field b with discrete theta angle pi/2 int q_{rho_1(b)} + (D_b phi)^2 + M^2 phi^2 - phi^4 where phi is a real scalar and it couples to b by the Z_2 symmetry phi->-phi For M^2>0 the Z_2 gauge field is Higgsed and it is the trivial phase For M^2<0 the theory flows to pure Z_2 gauge theory with topological action pi/2 int q_{rho_1(b)}, and thus it becomes the theory (2.6). Note the theory with phi is dual to free Majorana fermion, with M^2 mapped to the fermion mass, as reviewed in e.g. (2.9) of https://scipost.org/10.21468/SciPostPhys.7.1.007

---

## Round 1 · Referee Report · Anonymous (Referee 2) · 2021-11-19

Report

Warnings issued while processing user-supplied markup:

  • Inconsistency: Markdown and reStructuredText syntaxes are mixed. Markdown will be used.
    Add "#coerce:reST" or "#coerce:plain" as the first line of your text to force reStructuredText or no markup.
    You may also contact the helpdesk if the formatting is incorrect and you are unable to edit your text.

The paper discusses a certain higher-dimensional generalization of spin structure, where one extends the spacetime rotational/Lorentz symmetry group by a $\mathbb{Z}_2$ higher-form symmetry.

The paper is technical but would be of some interest to the people working within the field. As such, it can be published in the referee's opinion, after the following minor points are taken care of:

  • In footnote 2, why is the 1-form symmetry in this paper always unfaithful?

  • Above (2.11) and elsewhere, the authors seem to assume Kitaev's Majorana chain to be implicitly in the nontrivial phase; this implicit assumption is common in hep-th but not quite in cond-mat.

  • The notations used around (3.32) are rather confusing. That $\rho_2$ was used in (3.32) is OK, but the sentence immediately below explains $P(b)$ in terms of $q_{\rho_2}(b)$ saying that $\rho_2=0$. The referee needed a few seconds to understand that the author was not saying that $\rho_2$ should be set to zero in (3.32)...

  • In Sec. 3.6 and elsewhere, on non-spin manifolds, the discrete theta angle $p$ is valued in $\mathbb{Z}_4$, right? So it is not perfectly OK to distinguish $SO(3)_{\pm}$, which was OK on spin manifolds. At least some comments on this point would be necessary here.

  • In Appendix B, the authors should explain how to construct a quadratic refinement $q$ from the cochain $\rho_2$ satisfying $d\rho_2 = v$ where $v$ is the Wu class.

  • In footnote 46, is there actually a diffeomorphism for an arbitrary $a,b,c,d\in \mathbb{Z}$ with $ad-bc=1$?

========

There are some issues in English spelling and the grammar. The referee understands that young postdocs and graduate students do not have much time to polish the prose, but the authors should at least go through the manuscript to spot the immediate mistakes.

Here are some examples:

p.11: can be rephrases -> can be rephrased

p.11: on it boundary -> on its boundary

p.14: a sentence fragment "that only emerges in the low energy physics" remains. The authors seemed to have forgotten to erase it, since the same content appears in the previous sentence.

p.14: The title of the subsection 3.1.5 is in the imperative, which is very unusual. Extend -> Extending

p.15: The second time-reversal symmetry is discussed -> was discussed

p.15: loop that charged -> loop charged or loop that is charged

p.16: exotic properties that will be discussed in detail in Sec.3.1 .... Do the authors expect the reader to read the article in reverse? This sentence appears in Sec.3.2!

p.19: an valid background as $w_1(TM)$ -> a valid background as $w_1(TM)=0$

p.21: The first sentence is not complete.

p.23: This boundary chiral central is -> This boundary chiral central charge is

p.30: needs to attached -> needs to be attached

p.30: monople -> monopole (twice)

p.39: Begin with either ... -> Beginning with either ...

p.43: one can generalized -> one can generalize

p.46, footnote 42: space there the loops live -> space where the loops live

p.47: a $(n+1)$-dimensional -> an $(n+1)$-dimensional

p.47: The sentence leading to (B.1) starts with "the", which should be "The"

p.53: The sentence which includes (D.7) does not seem to be complete

Reference issues:

Some of the DOIs are repeated, e.g. [7] or [8]

Some of the titles are not correctly capitalized, e.g. [26], [34], [35], [63]

Somehow $\mathbb{Z}$ is $\digamma$ in [65]

---

## Round 1 · Referee Report · Anonymous (Referee 3) · 2021-11-20

Strengths

1-The paper is well-organized, reviewing the Kitaev model in 1+1 dimensions, then extending this analysis to 3+1 dimensions.
2-The paper is timely and pursues an interesting new direction: there has been great interest lately in the study of higher-group symmetries, and this paper considers a new example and application of such symmetries.
3-The application to fermionization is interesting.

Weaknesses

1-There are a number of spelling/grammatical mistakes, though none of them seriously hinder the reading of the paper.
2-The paper is rather focused on just a couple of examples, and it is unclear how general some of these phenomena will be.
3-Relatedly, since this paper deals with exotic symmetries, it is not applicable to Lorentz-invariant.

Report

The paper represents an interesting contribution to the literature on higher-group symmetries. However, some parts of it are unclear, and there are a number of grammatical errors (though, for the most part, they do not seriously hinder the reading of the manuscript). I recommend publication once some basic changes are made.

Requested changes

1-Non-causal reference: in section 3.2, you write, " This symmetry implies that the loops, which are charged under the one-form symmetry part of the two-group, have exotic properties that will be discussed in detail in Section 3.1." 2-Relatedly, it is never spelled out exactly what makes this loop "exotic." In particular, it seems to have nothing to do with the "exotic" symmetries studied recently by Seiberg and Shao in a series of papers. 3-Grammar, page 7: "the 0-form symmetry transforms disorder operator." 4-Third line of introduction should be "and" or "or" instead of "a" 5-p. 4: tense switches from present to past when discussing the appendix. 6-p. 10, just before equation (3.9): should say "the loops W transform" or "the loop W transforms" 7- p. 11, line 2: should say "can be rephrased" 8-p. 11, last paragraph of 3.1.1, "The spinor projective representation makes..." 9-p. 11, second line "on it boundary" doesn't make sense 10-The first sentence of the "Outlook" section is grammatically incorrect. 11-Finally, some equations seem to come out of thin air. (3.66), for example--where does the expression \int b \cup b come from? Either this should be explained in the text, or a reference should be given to a previous work that does explain this.

---

## Round 2 · Referee Report · Anonymous (Referee 3) · 2021-11-29

Report

The manuscript is now ready for publication.

---

## Round 2 · Referee Report · Anonymous (Referee 2) · 2021-11-30

Report

The revision is mostly satisfactory except for the addition to Appendix B.

There should be a mathematical theorem saying that a choice of the quadratic refinement is in 1-to-1 correspondence with the choice of the trivialization of the Wu structure. (For the simplest case of the Wu structure, i.e. the spin structure, this was done by Atiyah http://dx.doi.org/10.24033/asens.1205 using index theorem and then by Johnson http://dx.doi.org/10.1112/jlms/s2-22.2.365 more elementarily.)

As this is a physics paper, the authors do not have to explain it, but they at least have to provide a reference.
  • validity: -
  • significance: -
  • originality: -
  • clarity: -
  • formatting: -
  • grammar: -

Author:  Po-Shen Hsin  on 2021-12-01  [id 1997]

(in reply to Report 2 on 2021-11-30)

Thanks for the comment. The correspondence is explained in Ref. [35] (eg. Corollary 1.17).

Anonymous on 2021-12-27  [id 2056]

(in reply to Po-Shen Hsin on 2021-12-01 [id 1997])

I'm the referee and the authors were quite right, the point I raised was already in [35] which was already properly cited in v2 from Appendix B. I am thankful to the authors (and I am sorry for making them going through the trouble) to provide the new version v3 with an additional sentence in Appendix B to emphasize the correspondence between Wu structure and the quadratic function. I am also very sorry that I did not notice the authors' comment earlier and that my reply was very slow.

I think the v3 can be published as is.

---

## Round 2 · Referee Report · Anonymous (Referee 1) · 2021-12-4

Report

The authors have addressed satisfactorily most of the questions. I recommend the paper for publication in the present form.

---

## Round 2 · Author Response

We thank the referees and editors for their time and for thoroughly reviewing and improving our work.

---

## Round 2 · List of Changes

• The grammars and spellings mentioned in the reports are fixed, as well as the references (repeated DOI, capitalized titles, and math symbols).
  • p3 added clarification in the second paragraph that the 1+1d theory is the non-trivial phase of the Kitaev's chain.
  • p3 added clarification in footnote 2 about the terminology of unfaithful higher form symmetry i.e. symmetry generator invariant under small deformations of the submanifold where the generator is supported.
  • p7 added (2.7) and an explanation that the theory (2.6) is invertible i.e. gapped with a unique ground state. The explanation is referred to later in the paragraph below (3.21) and (4.4).
  • p16: below (3.22) correct the non-causal reference ``will be discussed in Section 3.1" -> "as we discussed in Section 3.1".
  • p19: below (3.32) added clarification about the Pontraygin square P and the quadratic function q.
  • p25: added footnote 25 using anti-semion as an example to explain the chiral central charge mentioned here.
  • p29 beginning of Section 3.6, added clarification that the SO(3)- theory discussed here has the discrete theta angle p=1 (as opposite to p=3).
  • p29 beginning of Section 3.6, added clarification that "m=3" stands for the Z_2 one-form symmetry SPT phase with the partition function (E.1) with m=3.
  • p33: in equation (3.66) added clarification about where b cup b comes from (difference of q(b) and -q(b)).
  • p34 figure 3 caption: added that the analogous 1+1d action for (3.68) is given by Z2 gauge theory+ Ising scalar as in (2.9) of Ref [55], and it is dual to free massless Majorana fermion, with the fermion mass identified with the mass square of the Ising scalar.
  • p35: in (3.68) changes the sign of lambda_{12}.
  • p35: in the bullet point M^2<0, lambda_{12} is replaced by lambda'.
  • p49: in Appendix B added a final paragraph about a construction of the quadratic function using the Wu3 structure.
  • p50 footnote 49: added that the general SL(2,Z) map is not a diffeomorphism, while the mapping class group is D8.

---

## Round 3 · List of Changes

added reference in the last paragraph of Appendix B.

---

## Editorial Decision

published